# A Scalable Transformer-based Framework for Fault Detection in Mission-Critical Systems

## Abstract

Detecting underlying faults is crucial in the development of mission-critical planning systems, such as UAV trajectory planning in Unmanned aircraft Traffic Management (UTM), which is vital to airspace safety. Inevitably, there exists a small set of rare, unpredictable conditions where the UTM could suffer from catastrophic failures. Most traditional fault detection approaches focus on achieving high coverage by random input exploitation. However, random methods are struggling to detect long-tail vulnerabilities with unacceptable time consumption. To tackle this challenge, we propose a scenario-oriented framework to search the long-tail conditions, accelerating the fault detection process. Inspired by in-context learning approaches, we leverage a Transformer-based policy model to capture the dynamics of the subject UTM system from the offline dataset for exploitation acceleration. We evaluate our approach over 700 hours in a massive-scale, industry-level simulation environment. Empirical results demonstrate that our approach achieves over 8 times more vulnerability discovery efficiency compared with traditional expert-guided random-walk exploitation, which showcases the potential of machine learning for fortifying mission-critical systems. Furthermore, we scale the model size to 2 billion parameters, achieving substantial performance gains over smaller models in offline and online evaluations, highlighting the scalability of our approach.

## 1 Introduction

Unmanned aircraft system Traffic Management (UTM) (Kopardekar, 2014; Kopardekar et al., 2016) is a mission-critical system to ensure safety and coordination in low-altitude aircraft operations. As Unmanned Aerial Vehicles (UAVs) are increasingly applied in civilian and commercial tasks, such as logistics (Yang Su et al., 2023; Chen et al., 2024), disaster relief (Kshitij Aggarwal & Aayush Goyal, 2021; Murat Bakirci & Muhammed Mirac Ozer, 2023), and environmental monitoring (Biruk E. Tegicho et al., 2023; Manilo Monaco et al., 2022), the vulnerability discovery of UTM systems is crucial to prevent accidents in complex, real-world scenarios (G. Raja et al., 2021; Wedad Alawad et al., 2023), which brings about the need for rigorous testing in the verification phase.

Testing in UTM is particularly challenging due to the long-tail effect of potential failure scenarios (Wang et al., 2022; Feng et al., 2023). In UTM, the majority of operational scenarios are safely managed by the system's self-healing design, while a small subset of rare and unpredictable situations can lead to severe safety risks and system-wide failures. These low-probability, high-risk scenarios are difficult to identify due to infrequent occurrence and obscurity within a vast space of relatively safe scenarios. However, traditional testing methods, such as random scenario injections (Zhong et al., 2021; Nicholas B. N. Nyakundi et al., 2023) and coverage-based techniques (Flood & Korenko, 2013; Nalic et al., 2020), struggle to detect rare critical edge cases efficiently. The random nature of these approaches leads to excessive exploration of low-risk situations, resulting in wasted resources and a low likelihood of exposing the most dangerous vulnerabilities in a timely manner.

To address this issue, we propose a novel search-oriented testing framework that treats fault detection as a search problem across the long-tail of operational scenarios. The framework consists of two main components: a Policy Model (PM) and an Action Sampler (AS). Drawing inspiration from in-

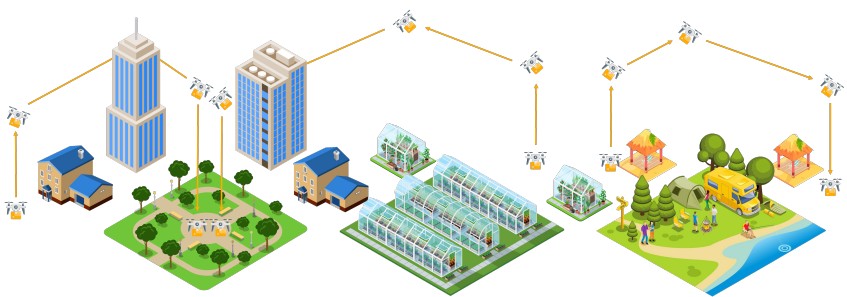

Figure 1: **Overview of the operational environment in Unmanned aircraft system Traffic Management (UTM) of System-Under-Test (SUT).** The UTM system operates in a variety of environments, including urban, suburban, and rural areas. Each setting poses distinct challenges, such as high-density air traffic in urban regions and limited infrastructure in rural areas, requiring strong management and coordination strategies. Rapid fault detection across these diverse scenarios is essential for maintaining safety and preventing catastrophic failures in real-world deployments.

context learning methods (Laskin et al., 2022; Fu et al., 2024; Benjamins et al., 2022), the PM uses a Transformer-based reinforcement learning approach to generate fault injections. It takes both historical data and real-time System-Under-Test (SUT) states as input, capturing temporal dependencies and operational dynamics. This design allows the model to identify patterns in similar operational dynamics and generalize to unseen environments. The PM can manipulate environmental factors (e.g., placing obstacles) and modify the internal states of drones (e.g., simulating poor network connectivity). Before being injected into the SUT, the testing scenarios generated by the PM are refined by the rule-based AS. The AS incorporates human preference alignment using logit bias (Tang et al., 2024; Brown, 2020) and prevents invalid actions through rejection sampling (Dubey et al., 2024). This ensures strict adherence to safety constraints and narrows the search space. We validated our approach in an industry-level UTM simulation environment over the course of 700 hours. The System Under Test (SUT) was a UTM scheduling platform that managed over 400 drones performing food delivery tasks across 30+ distinct environments, as shown in Fig. 1.

The results demonstrate that our framework accelerates vulnerability discovery by more than eight times compared to traditional methods, while also identifying critical scenarios that conventional techniques failed to detect. Additionally, we examined the effect of model scaling and found that increasing the parameter size to 2 billion significantly improved performance. This highlights the scalability and robustness of our approach in enhancing safety for mission-critical systems.

In summary, the key contributions of this paper are as follows:

- We introduce a novel testing framework that combines a Transformer-based policy model for scenario generation with a rule-based action sampler for targeted scenario testing in UTM systems. This integration significantly improves the efficiency of fault detection.

- We propose a Transformer-based offline reinforcement learning architecture that effectively captures multi-agent system dynamics and enables efficient exploration of long-tail scenarios.

- The effectiveness of our approach is thoroughly validated through extensive online simulations, spanning 700 hours across diverse real-world environments. The results show significantly better performance compared to traditional methods and human experts.

## 2 RELATED WORKS

**Multi-Agent System Testing** Multi-Agent Systems (MAS), such as autonomous vehicles (Li et al., 2023; Feng et al., 2023; Daniele et al., 2024; Ashwin & Naveen Raj, 2023) and robotic swarms (Xia et al., 2022; Mai et al., 2022; Lv et al., 2023), process continuous temporal information from both inter-agent interactions and agent-environment dynamics. As MAS complexity and importance grow, testing to identify potential vulnerabilities has gained increasing attention (Daniele et al., 2024). For instance, Kim et al. (2022) introduced an end-to-end mutational fuzzing method for

Autonomous Driving Systems (ADS), employing four parameter mutation strategies to explore the problem space from initial scenarios. Additionally, Yao et al. (2023) used graph centrality analysis to detect GPS spoofing vulnerabilities, while Li et al. (2023) applied the TD3 reinforcement learning algorithm to reduce randomness and inefficiency in fuzzing methods. Despite these advances, fuzzing methods often fail to balance coverage and efficiency, as much of the explored problem space offers limited value (Daniele et al., 2024). Traditional testing techniques, such as fuzzing (injecting random or semi-random inputs) (Miller et al., 1990; Zhang et al., 2021) and mutation testing (introducing controlled changes to the system) (Jia & Harman, 2011), have been adapted for MAS. Mutation testing may fail to capture the emergent behaviors arising from complex agent interactions, limiting quality and diversity (Daniele et al., 2024).

**Sequential Trajectory Modeling**   Early efforts primarily utilized transformers with self-attention mechanisms (Vaswani et al., 2017) as encoders to process complex information embedded within agent trajectories. These trajectories, modeled as sequences of tokens (Chen et al., 2021; Chebotar et al., 2023b; Wu et al., 2023), contain observations, interactions between agents, and feedback from the environment at different time steps. The pioneering work by Zambaldi et al. (2018) introduced the concept of using multi-head dot-product attention to capture relational reasoning over structured observations. This approach was later successfully implemented in AlphaStar (Vinyals et al., 2019) to manage multi-entity observations in the complex multi-agent game StarCraft II (Samvelyan et al., 2019). Trajectory Transformer (Janner et al., 2021) adopted sequence modeling techniques, such as beam search, for reinforcement learning tasks, aiming to mitigate the correlation and bias involved in jointly modeling states and actions. In complex and dynamic environments, agents' behaviors are heavily influenced by each other and their surroundings. AgentFormer (Yuan et al., 2021) addressed this by incorporating agent identifiers into the attention mechanism to model the influence of each agent's trajectory on others. Scene-LSTM (Manh & Alaghband, 2018) divided environments into grid cells, while Scene Transformer (Ngiam et al., 2022) aggregated environmental object context and agent interactions through attention layers to produce unified future state predictions.

**Scenario-Based Testing**   Scenario-based testing offers a more structured approach to evaluating MAS by analyzing the system's behavior under specific conditions. Tian et al. (2022) demonstrated the effectiveness of motif-driven paths in identifying distinct safety violations. However, manually crafting realistic and comprehensive scenarios is both time-consuming and challenging. To address this, learning-based methods have gained traction. Feng et al. (2023) formulated accidents in autonomous vehicles as a sequential Markov Decision Process (MDP), using a reinforcement learning framework, D2RL, to manipulate trajectories and simulate accidents by controlling nearby vehicles. Similarly, Tian et al. (2024) explored the use of Large Language Models (LLMs) in the LEADE technique to automate scenario generation. Additionally, Fu et al. (2024) introduced models that adapt to new tasks by leveraging current contextual information. Through in-context learning, models can use relevant examples or instructions to guide the generation of test scenarios or identify potential vulnerabilities in more targeted and adaptive ways (Wei et al., 2022).

## 3   FAULT DETECTION PROBLEM IN TESTING PHASE

In this section, we provide an overview of fault detection problem in testing phase of the UTM system. We first introduce the UTM system, specifying the importance of testing phase in the development of UTM system. After that, we analyze the testing framework for UTM system, including its role within development of UTM system, interactions with SUT, and targets of testing framework.

### 3.1   UTM SYSTEMS

**Targets of UTM Systems**   The UTM systems are designed for real-time or near-real-time organization, coordination, and management upon UAV swarms. UTMs generate control and command signals to UAV swarms for geo-fencing, route optimization, de-confliction, etc. (ICAO, 2023). The UTM system is usually developed as a complex system, integrating several modules. Since failures in UTM usually result in substantial economic losses or even human injury and death, UTM systems are considered as mission critical systems (Kopardekar, 2014) Therefore, UTM systems are required be of high fault-tolerance. This raises demands on rigorous fault detection in testing phase before real-world deployment. Detailed characteristics of UTM are available in Appendix A.1.

**Fault Detection in UTM Systems** Fault detection refers to the process of identifying potential failures (or faults) before deploying UTM in real-world services. This process is of significant importance to rule out possible failures in advance. Typically, testing phase contains several steps, with the complexity of testing scenarios increasing progressively. As reported by the Federal Aviation Administration (FAA), despite extensive testing, long-tail faults still threaten the safety of UTM systems (Rios et al., 2017; FAA, 2023). These long-tail faults are particularly challenging to detect because the complex scenarios in which they occur are difficult to generate. This difficulty is compounded by the self-healing capabilities of UTM systems, which prevent the testing framework from accessing or inducing these fault conditions. Motivated by this challenge, we propose a novel testing framework designed to accelerate fault detection and enable the discovery of new faults.

### 3.2 Functionality of Proposed UTM Testing Framework

The proposed testing framework operates as a testing module that runs alongside the UTM system, monitoring its behavior and injecting controlled disturbances to SUT.

**Input Stream** The testing framework continuously receives three types of real-time data from simulator: (1) **UAV Runtime Data**: Including position, velocity, and acceleration information for all active UAVs (2) **Mission Status**: Flight plans, current objectives, and completion status for each UAV Data streams fed to either testing framework or the UTM system are strictly identical, which are available for RL modeling as states in 4.1.

**Targets of Testing Framework** The testing framework for UTM systems is designed to generate adversarial scenarios that evaluate system-wide behavior rather than individual on-device drone states. While traditional approaches to UTM testing focus on component-level verification, our framework aims to uncover vulnerabilities that emerge from complex system-wide interactions and temporal dependencies. Given the vast state space of UTM systems, we adopt a more focused approach: generating a carefully curated set of high-risk test scenarios that are most likely to reveal critical system vulnerabilities. This strategic reduction in test cases allows for more efficient and targeted testing while maintaining comprehensive coverage of potential fault modes.

**Challenges of Testing Framework** In developing this targeted testing approach, we identified three fundamental challenges: (1) **Complex Temporal and Inter-agent Dependencies:** UTM systems involve intricate temporal dependencies and agent interactions. Testing must consider both long-term effects and multi-agent behaviors, as vulnerabilities often emerge from their combined impact rather than immediate or single-agent issues. (2) **Long-tail Effect in Fault Distribution:** Trivial errors are often corrected by self-healing mechanisms of UTM. Critical faults typically occur in rare edge cases (e.g. multiple simultaneous errors), making them difficult to detect through conventional testing. (3) **Environmental Consistency:** Generated test scenarios must balance between discovering edge cases and maintaining physical plausibility in realistic operational settings. Our framework addresses these challenges through a novel combination of reinforcement learning techniques and domain-specific constraints, as detailed in the following sections.

## 4 Testing Problem Analysis and Modeling

In this section, we first transform the complexities of testing into a manageable RL problem, with the objective of learning to identify and trigger the aforementioned faults. Additionally, we employ an offline RL strategy combined with contextual information to effectively address the dynamic and high-dimensional nature of UTM systems.

### 4.1 Towards Reinforcement Learning

To address the challenge of scenario quality, we adopt an RL framework to formulate generation. The RL formulation allows us to model the complex temporal dependencies inherent in UTM systems and to learn effective strategies for exploring the state-action space.

Formally, let $\mathcal{O}$ denotes the space of observable individual UAV information, $\mathcal{S}$ represents the state space of testing framework with the UTM system managing $N$ UAVs, $\mathcal{A}$ the set of possible actions

or injections, and $R : \mathcal{S} \times \mathcal{A} \to \mathbb{R}$ a reward function that quantifies the impact of actions on system safety and performance. The objective is to identify $\mathcal{T}$, a set of sequences of states and actions $\tau : \{(s_1, a_1), ..., (s_T, a_T)\}$ that maximize the cumulative reward $R_\tau = \sum_{i=1}^{T} r(s_i, a_i)$, where high rewards correspond to the discovery of critical failure modes. In following paragraphs, we define state, action, and reward in UTM testing separately.

**State**   The state space of testing framework is a concatenate of UAV information in UAV numbers and time-steps. For information of individual UAV, the state contains both temporal, spatial information and runtime mission status. We define $o_i^t \in \mathcal{O} \subset \mathbb{R}^d$ as a vector of $d$ relevant features, which encapsulates the observable information for $i$-th UAV at $t$-th time-step, including: kinetic information (position, velocity, and acceleration of all UAVs), environmental data (obstacles, weather conditions, and airspace restrictions) and mission-specific details (battery levels, payload capacity, and route destinations). To capture both cross-UAV dependencies and temporal dependencies, we represent the state $s \in \mathcal{S}$ for testing framework as a sequence of observations of all $N$ UAVs over a fixed time window $T$, namely, $s = \{(o_1^1, ..., o_N^1), ..., (o_1^T, ..., o_N^T)\}$.

**Action**   The action space $\mathcal{A}$ comprises a discrete set of all possible injection operations, each targeting a specific component or aspect of the SUT. We define $\mathcal{A}$ as the Cartesian product of two sets $\mathcal{A} = \mathcal{D} \times \mathcal{F}$ where $\mathcal{D}$ represents the set of targetable UAV, and $\mathcal{F}$ is the set of $m$ applicable disturbance injections. For each injection, all the possible types are listed in Table 7 in Appendix A.7. Each action $a \in \mathcal{A}$ is a tuple $(d, f)$, where $d \in \mathcal{D}$ and $f \in \mathcal{F}$. This formulation allows for a combinatorial exploration of fault scenarios while maintaining a structured action space of cardinality $|\mathcal{A}| = |\mathcal{D}| \times |\mathcal{F}|$.

**Reward**   The reward function $R$ is designed to capture the system's safety and operational efficiency as $r(s_t, a_t) = \sum_{i=1}^{K} \alpha_i r_i(s_t, a_t)$ denoting reward at timestep $t$ where $r_i$ are individual reward components (e.g., collision avoidance, mission completion, system stability) and $\alpha_i$ are their respective weights.

## 4.2   OFFLINE REINFORCEMENT LEARNING

Considering sample inefficiency of traditional RL in complex environments, we raise a novel offline RL approach to improve. This methodology leverages a large, pre-collected dataset of UTM system trajectories, denoted as $\mathcal{T} = \{(s, R, a, r) \mid s \in \mathcal{S}, a \in \mathcal{A}, r, R \in \mathbb{R}\}$, where $s$ is the current state, $a$ is the action taken, $r$ is the immediate reward received at current timestep and $R$ is the return-to-go indicating potential reward in future steps. Thus the objective of problem is formulated as searching $\pi_\theta = \arg\max_\pi \mathbb{E}_{(s,R,a,r) \sim \mathcal{T}}[\Sigma r^\pi(s, a)]$ where $\theta$ are the parameters of the policy network.

**Transformer**   In order to tackle long-range dependencies in the trajectory data, as challenges described in Section 3.2, we employ a Transformer-based architecture motivated by the Transformer's ability (Radford et al., 2019) to model complex temporal relationships. Self-attention mechanism also provides interleaving data utilization among different head in favor of modeling agent-wise interaction. Our strategy of Transformer usage resides in (1) modeling complex system mechanism through learning reward/return, (2) generating targeted and valuable actions based on knowledge of world. Formally, given a sequence of state-return pairs $(s_1, R_1), ..., (s_T, R_T)$, the decoder-only Transformer processed this information end-to-end into a set of predictions $\{(\hat{R}_1, \hat{a}_1), ..., (\hat{R}_T, \hat{a}_T)\}$ with $\hat{R}$ as regressive modeling of world and $\hat{a}$ decision of actions, where $\hat{R}_t = f_\theta(s_1, R_1, ..., s_t)$ and $\hat{a}_t = f_\theta(s_1, R_1, ..., s_t, R_t)$ with $f_\theta$ denoting the Transformer decoding function with parameters $\theta$. This architecture enables the model to learn subtle patterns across extended time horizons, potentially uncovering intricate failure modes that might be overlooked by methods with more limited temporal reasoning capabilities.

**Context-Aware**   We incorporate context-aware scenario generation to enhance the generalization capabilities of our model. Our approach draws inspiration from recent advancements in in-context learning (Brown & Mann, 2020). Let $\mathcal{C}$ denote a context set comprising a small number of relevant historical trajectories. We augment our state representation to include this context $\tilde{s} = [\mathcal{C}; s]$ where $[\cdot; \cdot]$ denotes concatenation. This context-aware formulation allows the Transformer model to

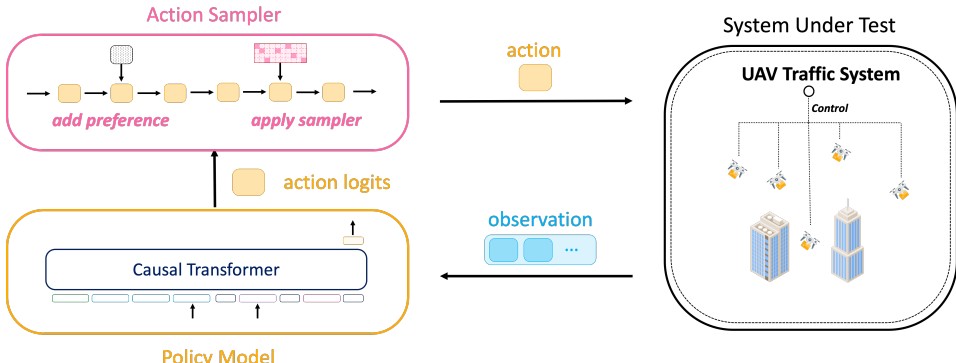

Figure 2: **Architecture overview of the proposed scenario-oriented testing framework.** The framework consists of two primary modules: (1) a Transformer-based Policy Model (PM) for generating fault scenarios based on real-time and historical SUT data, and (2) an Action Sampler (AS) that enforces predefined safety rules and filters out undesirable actions. The validated scenarios are then injected into the System-Under-Test (SUT) for evaluation. This architecture effectively narrows the search space to high-risk scenarios, improving fault detection efficiency and reducing unnecessary exploration of low-risk cases.

utilize longer-ranged data and adapt its behavior based on relevant historical examples, potentially improving its decision performance in novel or underrepresented scenarios due to self-healing UTM functionality to automatically resolve disturbances.

**Action Sampling with Domain Constraints**    To ensure the physical plausibility of generated scenarios, we introduce a constrained action sampling mechanism. Let $\Phi(s)$ represent a set of domain-specific constraints that define the feasible action space given the current state $s$. We modify the action selection process as $a \sim \pi(a|s) \cdot \mathbb{1}[a \in \Phi(s)]$ where $\pi(a|s)$ is the learned policy, and $\mathbb{1}[\cdot]$ is the indicator function. This indicator only functions during inference to allow for the incorporation of expert knowledge and system-specific constraints without compromising neither learning efficiency nor the learned policy's flexibility. In training, we added a stage of prediction for *available action mask* which serves to aid regression in system modeling. Action predictions are used directly in training or through sampler in inference.

## 5 METHODOLOGY

In this section, we introduce an automatic framework with generative capability for complicated scenarios and interface for prior preference alignment and knowledge accumulation. As shown in Fig. 2, in this framework, we utilize a Transformer-based model as policy model (PM), initiating the process by producing a set of actions based on the system state. According to action space defined in Section 4.1, generated actions can be interpreted as potential fault injections to be applied to typical victim drones in UTM. Subsequently, these actions are passed through a domain-specific action sampler (AS). AS serves for two purposes: (1) ensure the PM-generated actions available within the specific UTM context; (2) leverage human expert knowledge to re-sample actions with balanced preference bias in chosen actions and agents. Only actions sampled are injected into the system-under-test (SUT). On SUT finishing execution, a new system state would be generated and fed back to the PM, along with the evaluation of the actions (reward). Thus PM continuously refine scenarios to uncover potential vulnerabilities.

### 5.1 POLICY MODEL

In this subsection, we describe the design of PM, according to RL formulation defined in 4.1. The Policy Model serves as the generative engine of our framework, leveraging the power of Transformer architectures to capture complex temporal dependencies and system dynamics. During training, PM serves to model trajectory sequence from UTM and learn internal natures in offline dataset. In

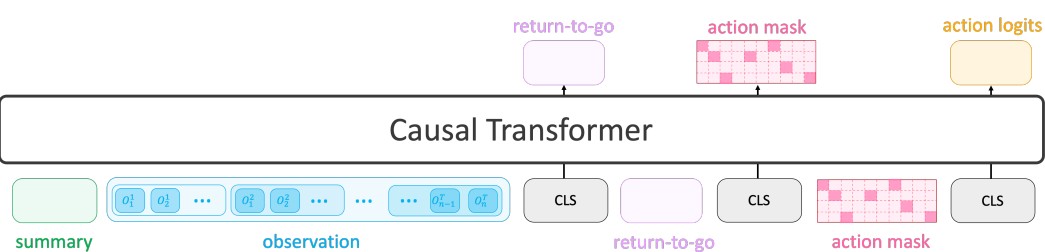

Figure 3: **Architecture of the Policy Model (PM).** The PM utilizes a Transformer-based reinforcement learning framework, taking both historical and real-time SUT states as input tokens to capture temporal dependencies and system dynamics. The model generates action sequences that include both environmental manipulations (e.g., placing obstacles) and internal state changes (e.g., network degradation).

performing inference, PM processes real-time UTM context and generates proposed fault injection actions to AS.

**Time sequence and action modeling** Expanding on previous work that utilized Transformers for decision-making (Chen et al., 2021), we design a unified time sequence format where observations $o$, actions $a$, returns $R$ and rewards $r$ of each agent are embedded to a homogeneous space after linear projections. Original rewards $r$ serve to construct *summary* tokens by aggregating increments in last $T$ timesteps, similar to the construction of return-to-go token $R$ (summarizing $T$ incoming timesteps). Input sequence thus carries data of in-total $3 \times T$ timesteps while focusing on central $T$ *current* timesteps. Tokens would then be arranged as array of $\langle \mathbf{O}, \mathbf{R}, \mathbf{A} \rangle$ tuples with length of $T$ timesteps. Considering temporal dependency in decision making, we masked out $\mathbf{R}$ and $\mathbf{A}$ tokens except that in last time step. Thus model utilize $T \times N$ observation tokens to predict the current return-to-go token $\hat{R}$ to fit ground-truth return $R$, as learning of implicit system nature. An intermediate *mask* token is introduce to mask out invalid action choices, in favor of modeling system capability according to current state.

**Embedding and Causality** To enhance the modeling of causal dependencies within the policy model, we employ a multi-faceted approach. We augment the sequentially sampled multi-agent drone observation data with positional embedding. Additionally, as shown in Fig. 3, input sequence is augmented with different classification (CLS) tokens as powerful discriminators in order to reduce the ambiguity of prediction targets. Inspired by insights from Shaw et al. (2018), we prioritize the most recent observations by placing them closest to the CLS token, ensuring that the model pays particular attention to the latest information when making decisions. This aligns with the principle that recent events often carry more causal relevance than distant ones.

To capture long-range dependencies, we employed self-attention mechanism among tokens together with a semi-lower-triangular agent-wise causal mask in attention calculation to preserve decision causality. Observation tokens $o$ at identical timestep are visible to each other homogeneously. However the $\hat{R}$ tokens could be predicted with only *observation* tokens visible before being fed with ground-truth *return-to-go* token. And only older $\langle \mathbf{O}, \mathbf{R}, \mathbf{A} \rangle$ tuples are visible to newer ones. We aim to guide the model to construct a more comprehensive and nuanced understanding of the causal dynamics. Formally, we can sequentially express the prediction task as $\hat{a}_t = f_\theta(S_{-t:1}, o_{1:t}, R_t, M_t)$ where $S$ denotes the *summary* token aggregating previous $T$ time steps and $f_\theta$ represents the Transformer model with parameters $\theta$.

### 5.2 ACTION SAMPLER

Inductive bias and generality are key drawbacks of traditional offline RL methods. We design a set of sampling strategies as a workaround. In this subsection, We first introduce preference bias as a notation of human feedback. And we describe action sampler functions between PM and SUT.

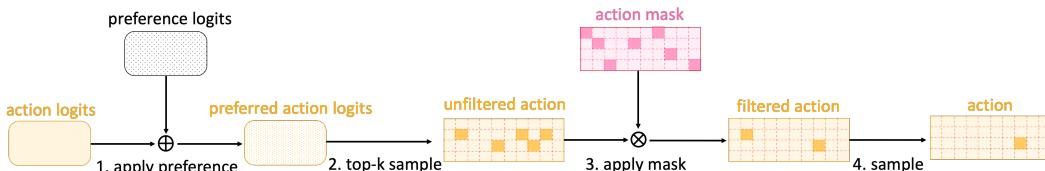

Figure 4: **Pipeline of the Action Sampler (AS).** The AS enforces safety constraints and domain-specific rules, filtering out irrelevant actions generated by the Policy Model (PM) before injecting them into the System-Under-Test (SUT), ensuring the integrity of the testing process.

**Preference Bias**   During the training process of decision-making models using auto-regressive models such as offline reinforcement learning, there is usually an uneven distribution of the output due to the collected training data, with little chance of sampling low-frequency choices. Meanwhile, the more complex the system is tested, the more insidious the vulnerability and the more significant the long-tail effect. In this work, training dataset is collected through traditional stress testing, where unpredictable inductive bias is common in production systems.

We introduce *Preference Bias*, improved from popularity bias (Klimashevskaia et al., 2024) with additional domain expert knowledge, to unify imbalance in model prediction and gap in prior human preference. Preference bias carries a expected distribution of $\langle \text{UAV}, \text{Action} \rangle$ tuples. The output of the offline-trained PM is augmented with compensation dynamically calculated from distance between recent historical trajectories and given distribution.

**Action Candidate Sampling**   As shown in Fig. 4, action logits predicted by PM are compensated according to preference distribution. To address long-tail effect and improve fairness (Menon et al., 2020), Top-K sampling is introduced after augmentation in order to maintain variance. Considering realistic capability of system status, immediate action mask is applied in order to filter intolerable action candidates. The final action is sampled through a uniform sampling after masking. By combining the generative power of the Transformer-based Policy Model with the refined selection process of the Action Sampler, our framework achieves a balance between exploration of complex failure scenarios and adherence to real-world constraints. This approach enables more efficient and effective testing of UTM systems, potentially uncovering critical vulnerabilities that traditional methods might miss. In below sections, we illustrate our advantages through experiment results.

## 6  RESULTS

We train the proposed framework with a large-scale offline dataset of around 17B tokens collected from stress testing data and evaluate on an industry-level simulator. As is summarised in Table. 8 in Appendix A.9, the training set consists of seven distinct regions and online testing includes two regions. The training dataset covering diverse geographical and operational characteristics, including a mix of rural (12.2%), suburban (39.0%), and urban areas (48.8%), each with varying numbers of UAVs, airports, and flight lines. The dataset is balanced to represent the typical distribution of scenarios encountered in real-world UTM systems. For testing, two regions (TR1 and TR2) are excluded from the training set to provide evaluations of the generalization capabilities.

We design two model of different size, with 1.2 billion and 2 billion parameters (referred as PM-1.2B and PM-2B respectively). We train each model on 16 NVIDIA A100 GPUs, each equipped with 80GB of memory. The training utilized PyTorch's Distributed Data Parallel (DDP) to efficiently distribute the workload across multiple GPUs, ensuring high computational efficiency and resource utilization. During training, the dataset is divided into smaller slices of 3B tokens for sequential loading during training.

We evaluate the performance of the proposed model through both offline and online evaluations to provide a comprehensive analysis. In Section 6.1, we focus on the offline evaluation of the PM's behavior during training, where we analyze the evolution of action accuracy and return-to-go loss. In Section 6.2, the online evaluation measures the model's performance in a deployed real-world environment, where we collect and analyze a range of key metrics. This dual evaluation framework offers a holistic view of the model's efficacy, ensuring robustness both during training and in practical applications.

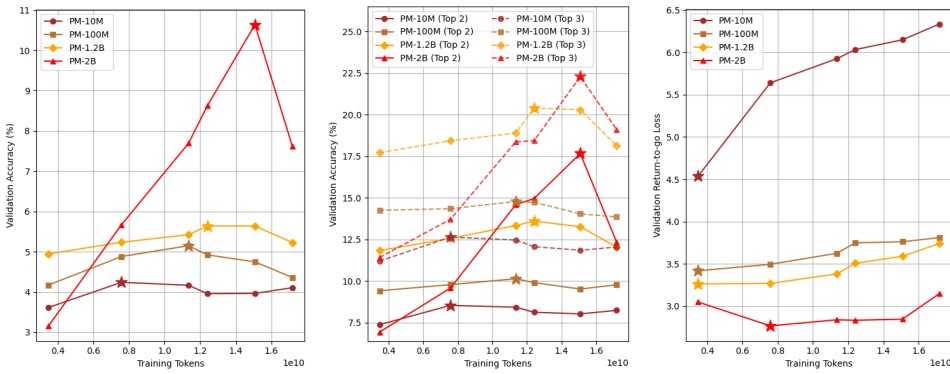

(a) Action accuracy of PM; (b) Top 2/3 action accuracy; (c) Return-to-go loss of PM;

Figure 5: **Offline evaluation results on validation sets during training.** The action accuracy and return-to-go of the models (PM-10M, PM-100M, and PM-2B) measured over increasing training tokens on validation sets. All models show an initial increase in accuracy, followed by a decline, indicating overfitting phenomenon. Similarly, all models eventually increase in return-to-go loss, signaling overfitting. Larger models demonstrate a clear advantage, achieving significantly higher accuracy lower return-to-go loss compared to the smaller models. The peak action accuracy for each curve is highlighted with a star.

| Category | Purpose | Metric |
|---|---|---|
| Action Probability | Measure the preference of framework. | Action Probability per Observation (APO) Action Probability Distribution (APD) |
| Action Quality | Evaluate the quality of generated actions. | Hazard Action Ratio (HAR) Constant-Pressure Action Ratio (CAR) |
| Testing Efficiency | Evaluate the effectiveness of framework. | High Risk Scenarios per Million Flights (SPM) Faults per Million Flights (FPM) |

Table 1: **Metrics for online evaluation of testing performance.** The metrics are categorized into three groups for a comprehensive evaluation of the proposed testing framework's capabilities, including the preference and quality of proposed framework, as well as the final results. The detail definition of metrics can be found in Appendix A.6.

## 6.1 OFFLINE EVALUATION

For offline evaluation, we focus on the impact of model size on action accuracy and return-to-go loss during training. Especially, we apply the top K action accuracy in that in our framework, actions are sampled based on the top-k predictions rather than solely the top-1. The results in Fig. 5 illustrate that larger models consistently perform better across both action accuracy (highest) and return-to-go loss (lowest) metrics. This indicates that larger models have a better capacity to capture the underlying structure in the offline data, achieving more accurate action selections with fewer training tokens. Fig. 5 also reveals that the PM-2B model begins to overfit much later compared to the smaller PM-10M and PM-100M models. This suggests that larger models not only perform better in terms of action accuracy but also exhibit better generalization properties, allowing them to continue learning effectively with more data before encountering overfitting issues. This behavior is a hallmark of the scaling effect, where larger models benefit from increased capacity and more robust training dynamics, making them more resistant to overfitting compared to smaller models.

## 6.2 ONLINE EVALUATION

To evaluate the effectiveness of proposed framework in unseen environments, we evaluate our We selected several key metrics to evaluate the preference and effectiveness of PM, as well as the quality of actions, as is shown in Table. 1. For detailed explanation of each metric, we refer to the Appendix A.6.

From the results shown in Table. 2, we can conclude that the proposed PM-2B model significantly outperformed both expert-guided testing and smoke test baselines across all key metrics.

| Metrics | PM-2B | | PM-1.2B | | Expert-Guided Exploitation | | Smoke Test* | |
|---|---|---|---|---|---|---|---|---|
| | TR1 | TR2 | TR1 | TR2 | TR1 | TR2 | TR1 | TR2 |
| APO(%) | **20.0** | **31.5** | 55.3 | 38.3 | 72.0 | 83.3 | 100 | 100 |
| APD(%) | 26/34/21/19 | 46/32/11/11 | 28/27/22/23 | 30/29/20/21 | **25/25/25/25** | **25/25/25/25** | N/A | N/A |
| HAR(%) | **10.8** | **4.9** | 6.7 | 4.2 | 3.6 | 1.7 | N/A | N/A |
| CAR(%) | **29.7** | **64.1** | 4.0 | 4.5 | 4.1 | 3.9 | N/A | N/A |
| SPM | **50.5** | | 17.6 | | 5.8 | | N/A | |
| FPM | **7.6** | | 2.2 | | <1.0** | | <1.0** | |

Table 2: **Performance metrics of the propose framework in online environments of unseen regions.** This table shows the online results in out-of-distribution region TR1 and TR2. Results of PM models are reported on over 700 hours testing in total, with around 100M records for each model in each region. The detailed definition of metrics can be found in Table. 1 and Appendix A.6.
*: The smoke testing refers to the basic functionality testing of UTM system. This is conducted as the initial testing after a new build or version of the UTM system.
**: The FPMs are below 1.0 because the two baseline tests have already been thoroughly used to identify existing bugs and improve UTM in advance, while our method is focused on discovering new bugs in the updated version of the UTM system after the baselines have reached their detection limits.

Specifically, PM-2B generates high-risk scenarios weight times faster than smoke testing, and is able to discover bugs while expert-guided testing method fails to. This indicates that the proposed framework is more effective in identifying critical scenarios and potential failures. Furthermore, comparing with smaller PM-1.2B model, PM-2B performs significantly better in action quality and efficiency. This suggests the existence of scaling effect between model size and online performance in discovering critical cases and efficiently covering high-risk regions. Interestingly, the PM-2B model detected failure modes (SPM and FPM) that the smoke test completely missed. This emergent capability shows that the PM framework can find faults beyond traditional rule-based methods, demonstrating its utility for uncovering rare bugs. Considering both the scaling effect and emergent abilities, our framework shows significant promise for scaling up model sizes, and has the potential to become a breakthrough (akin to a "ChatGPT-moment") in the testing field in the future. However, PM models fail to balance the distribution of different action types, which could lead to potential under-exploration in less frequent action spaces. This suggests a need for better action sampling strategies.

**Why does proposed framework exceed the performance of human experts?** Although trained with expert-guided exploitation data, PM model ultimately surpass the performance of human experts. This is attributed to that PM model applies offline RL, which can be viewed as an implicit filter of low-quality actions (Prudencio et al., 2023), making it less susceptible to distraction during the search for long-tail scenarios.

We can illustrate this by analyzing the hazard action ratio per observation, which is obtained by multiplying HAR and APO, and the constant-pressure action ratio per observation, calculated by multiplying CAR and APO. For both PM-2B, PM-1.2B, and human experts, the hazard action ratio per observation is consistently around 2%. This shows that all methods are similarly effective in identifying high-risk actions. However, the key difference is that the PM models demonstrate a significantly higher constant-pressure action ratio per observation, indicating that they maintain a more sustained level of high-risk actions over time. This ability to constantly pose challenges and maintain pressure highlights the advantage of the PM models in exploring complex, high-risk scenarios more thoroughly, thereby leading to superior fault detection and scenario coverage.

## 7 CONCLUSION

We propose a novel scenario-oriented testing framework for identifying vulnerabilities in mission-critical systems, specifically applied to UTM. Our approach leverages a Transformer-based policy model to tackle long-tail effect and efficiency challenge in fault detection. Context utilization in policy model improves generality in unseen regions. Our results highlight the potential of learning and expert hybrid approaches in fortifying mission-critical systems. The end-to-end auto-regressive learning methodologies are worth studying. Future work could explore the application of this framework to other mission-critical domains beyond UTM, such as autonomous vehicles or industrial control systems.

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

## A  APPENDIX

### A.1  UTM SYSTEM ARCHITECTURE AND TESTING PIPELINE

**What is Unmanned aircraft Traffic Management (UTM) system?**    The Unmanned aircraft Traffic Management (UTM) system, as introduced by the National Aeronautics and Space Administration (NASA) (Kopardekar, 2014; Kopardekar et al., 2016), is designed to ensure safe and efficient operation of multiple unmanned Unmanned Aerial Vehicles (UAVs) in shared airspace. The UTM concept is developed to support the integration of UAVs into airspace without requiring human air traffic controllers to manage every UAV directly. Instead, UTM emphasizes the use of automated systems to coordinate UAV operations. This includes services like geofencing, route optimization, and deconfliction, ensuring that UAVs can safely and autonomously operate in both sparsely populated rural and densely populated urban areas or alongside manned aircraft.

UTM is typically developed as a complex system. This is because the UTM systems should integrate a wide range of functionalities and address diverse challenges associated with managing UAV operations in dynamic and unpredictable environments. UTM systems need to handle real-time communication between UAVs, ground stations, and other stakeholders, while simultaneously ensuring safety, efficiency, and fairness in airspace usage.

As show in Figure 6, UTM serves as the central coordinator, processing dynamic information received from all UAVs and managing overall traffic flow through sophisticated decision-making algorithms simultaneously. UTM maintains continuous communication, flight route allocation and trajectory assignment with multiple UAVs, each equipped with various sensors and control systems, while simultaneously monitoring environmental conditions and potential conflicts.

**What is fault detection in development of UTM and why it is important?**    We define the term *fault detection* as the process identifying possible faults in the UTM system during testing phase, which is before the UTM system is deployed in real-world environments. It is typically divided into several steps, including module testing, integration testing, smoke testing (functional testing), stress testing, etc. After each testing step, the confidence (e.g., reliability, fault tolerance, and compliance with regulatory standards) of UTM system increases as potential faults are identified and addressed, ensuring that the system becomes progressively more robust and reliable.

Fault detection is a critical aspect of UTM development because it directly impacts the safety, reliability, and efficiency of development pipeline. As a mission critical system, the UTM system should be designed to eliminate all the faults it may occur, which are usually costly or even deadly (e.g., UAV crashes, collisions with buildings or even collisions with human injuries) (Kopardekar, 2014; Kopardekar et al., 2016). By identifying and addressing potential faults during the testing phase, fault detection ensures that the UTM system operates as intended, mitigating risks before deployment in real-world environments. This proactive approach prevents costly failures, enhances system robustness, and builds trust among stakeholders.

**Why fault detection is challenging?**    Fault detection in UTM systems is inherently challenging, particularly as testing progresses through advanced stages. While early testing steps may uncover obvious issues, the long-tail of rare and hard-to-detect faults often remains persistent and elusive. This difficulty is compounded by the self-healing capabilities of modern UTM systems, which can mask subtle issues that may only emerge under specific conditions. As is listed in the Table 3, although several testing steps have been conducted, there still remains faults to threat the safety of the UTM system (e.g. shakedown effects found by Federal Aviation Administration in field testing) (Rios et al., 2017; FAA, 2023). Based on the stepwise testing and field testing results, we estimate the faults found in different steps of testing, as listed in Table 3. From data in the table, we can see that as several testing steps are conducted, there still exists faults to be detected, which is fatal in mission critical systems.

### A.2  PROPOSED TESTING FRAMEWORK

**Testing Framework**    Testing framework introduced in this work serves as a copilot with UTM, rather than deploying on individual UAV. It monitors identical data streams along with UTM, including UAV telemetry (position, velocity, mission status) and system state information. The UTM

| Fault Types | Module Testing | Integration Testing | Smoke Testing | Stress Testing | Fault Remaining |
|---|---|---|---|---|---|
| Module Level | $\sim 20\%$ | $\sim 10\%$ | $\sim 30\%$ | $\sim 40\%$ | $\sim 0.1\%$ |
| Interface Level | $\sim 10\%$ | $\sim 20\%$ | $\sim 30\%$ | $\sim 40\%$ | $\sim 0.1\%$ |
| Running time | $\sim 10\%$ | $\sim 10\%$ | $\sim 40\%$ | $\sim 40\%$ | $\sim 0.1\%$ |
| **Scenario Complexity** | Simple | Simple | Medium | Medium | High |

Table 3: **Fault Types Detection during Different Steps of Testing.** The module testing verifies individual components of UTM to ensure they function correctly in isolation. The integration testing checks interactions between combined modules to detect interface issues. The smoke testing ensures basic functionality works correctly after a new build or update, acting as a preliminary check. The stress testing evaluates system stability and performance under extreme or peak load conditions. The tested scenarios for moduel testing and integration testing are relatively simple, while smoke testing and stress testing will generate more complex testing scenarios. As the testing steps conducted one by one, the software maturity of UTM increases gradually. However, there still exists rare faults happening in complex scenarios.

system provides trajectory schedule in favor of system robustness, while testing system generating adversarial disturbance actions to increase systematic vulnerability.

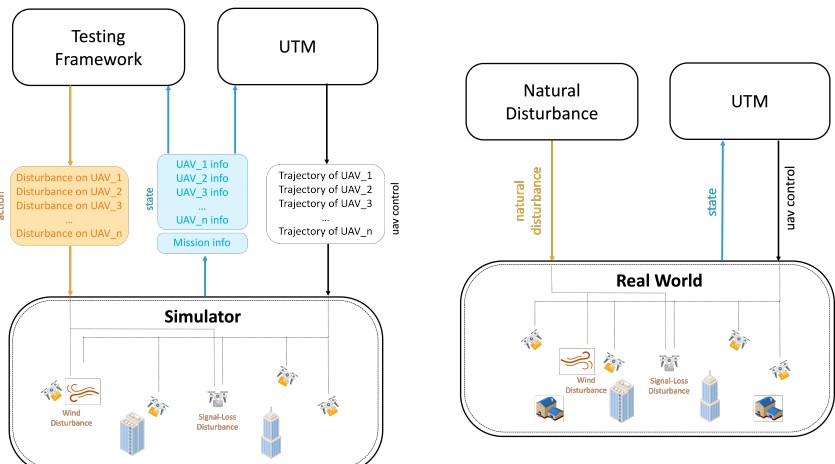

Figure 6: **UTM System and Testing Framework Architecture.** The testing framework works as copilot of UTM and operates on the server-side. As a mission critical system, UTM under test is designed as centralized architecture at once to insure the safety and remove potential conflicts in advance (Spalas, 2024; Hamissi & Dhraief, 2023). To align with the design of UTM, our proposed testing framework is also designed centrally. The testing framework mimics the natural disturbance to generate different scenarios.

Testing system is designed to manipulate external disturbances to UAVs like wind, obstacle and network jitter as shown in Table 7. Internal functionality and and robustness of on-device system of individual UAV is out of the scope of this research.

**Sim vs Real**   The framework's methodology emphasizes systematic exploration of edge cases and rare failure modes that might otherwise remain undiscovered in conventional testing approaches. Environmental disturbances suffer from randomness and difficulty in interpreting. In this work, we make use of simulators which enables configurable environmental disturbances and concrete mapping between them and consequential operating status, in favor of typical analysis and diagnosis. Visibility and capability of UTMs are strictly aligned in whether simulated or realistic context.

Besides, precise timing selection of disturbance injections is within consideration as well. Traffic pressure of UTM for complicated UAV MASs varies with time. Testing system learns to inject actions when UTM is handling the most vulnerable cases in favor of significance of tesing scenarios generated.

## A.3   CHALLENGES OF TESTING UTM

**Critical Fault Distribution Imbalance**   While UTM's fault-tolerant design successfully handles most anomalies through automated recovery mechanisms and redundant control strategies, this architectural resilience paradoxically increases the complexity of identifying severe failure scenarios, as intermediate failure states are often automatically corrected before they can develop into observable system failures. Critical failures, those capable of overwhelming the system's self-healing mechanisms, occupy an extremely small portion of the state-action space, which often reside in narrowly defined regions of the state-action space, requiring precise combinations of multiple adverse factors to overcome the system's multi-layer safety functionality. These regions are characterized by specific configurations of multiple elements: particular spatial arrangements of UAVs, precise timing of control actions, specific environmental conditions. Furthermore, these failure scenarios often represent emergent behaviors arising from subtle interactions between multiple system components and their recovery attempts, rather than simple violations of individual safety constraints.

| Types | Number of Influenced UAVs | Disturbance Times within 60s | Case Example | Real-World Ratio | Complexity |
|-------|---------------------------|------------------------------|--------------|------------------|------------|
| Safe Flight | 0 | 0 | N/A | $\sim 94\%$ | Low |
| Disturbances | 1 | 1 | Winds with exceeding magnitude | $\sim 5\%$ | Medium |
| | $\geq 2$ | 1 (each) | Winds hit multiple UAVs | $\sim 1\%$ | Medium |
| | 1 | $\geq 2$ | Winds hit twice with 60s interval | $\sim 0.1\%$ | High |
| | 1 | $\geq 2$ (simultaneously) | Signal Loss when Winds hit | $\sim 0.01\%$ | High |

Table 4: **Real-World UTM failure distribution.** In real-world UAV fleets, advanced UTM provides fundamental guarantee for safe flight, where faults with increasing risk still exist at a relatively low ratio and are increasingly hard to locate and tackle.

**High-Dimensional State-Action Temporal Dependency**   Testing of UTM systems confronts a fundamental challenge in navigating its inherent high-dimensional state-action coupling relationships. The state space encompasses multiple critical dimensions: spatial coordinates and velocity vectors of each UAV, environmental conditions, and communication network states. Each additional UAV exponentially expands this state space, creating a combinatorial explosion in the dimensions that must be considered during testing. Unlike traditional control systems where failures often manifest through immediate state violations, UTM system failures additionally emerge from specific combinations of historical state sequences and multi-agent coupling, as shown in Table 4. The behavioral trajectory of each UAV is intrinsically influenced by both its historical states and the temporal evolution of other agents' states in the shared airspace. For instance, a seemingly safe trajectory adjustment by one UAV could create cascading effects leading to system-wide conflicts minutes later through complex agent interactions. Furthermore, subtle perturbations in early states can propagate through the system's temporal dynamics to trigger critical failures in significantly later stages. The challenge is particularly pronounced in scenarios involving dense multi-UAV operations, where system behavior emerges from the intricate interplay of multiple agents' temporal trajectories rather than simple state-transition patterns.

## A.4   MOTIVATION FOR TRANSFORMER AND COMPARISON WITH OTHER MODELS

The main motivation of applying Transformer as backbone model lies in that the Transformer models are proved to be scalable in multi tasks (e.g., natural language processing (Brown, 2020), computational vision (Pan et al., 2021), robotics (Chebotar et al., 2023a), etc.). The scalability is of essential importance in the development of testing framework in that (1) complex temporal and inter-agent dependencies with scalable sizes of UAV swarm and temporal context window, and (2) long-tail effect in fault distribution requiring sufficiently large dataset to identify faults and to feed in backbone models. Leveraging the Transformer's inherent scalability in modeling extended context lengths and processing large-scale data inputs, it can effectively model complex temporal sequences and inter-agent interactions within UAV swarms of varying sizes. This capability allows the testing framework to accommodate extensive datasets necessary for identifying rare faults due to the long-tail effect in fault distribution. Furthermore, the Transformer's ability to handle large-scale data inputs ensures that the model remains robust and accurate as the system under test evolves (e.g. different region settings, as demonstrated in Table 5). Consequently, integrating the Transformer as the backbone

model enhances the framework's capacity to detect, analyze, and predict system behaviors across diverse operational scenarios.

However, alternative backbone models such as Graph Neural Networks (GNNs), Recurrent Neural Networks (RNNs), Long Short-Term Memory networks (LSTMs), and online reinforcement learning algorithms like Deep Q-Networks (DQNs) or Proximal Policy Optimization (PPO) often struggle to address aforementioned challenges effectively. These models may lack the inherent ability to capture long-range dependencies or scale efficiently with increasing sequence lengths and swarm sizes. Specifically,

- RNN/LSTM: RNNs and LSTMs encounter difficulties when modeling long temporal contexts due to issues like vanishing gradients, which add to the training difficulty. What's more, RNNs and LSTMs are hard to parallelized, which adds to the training time, especially when deal with large datasets (Devlin, 2018). Base on our primely experiments, we find that for models below 10 million parameters, RNNs are 10 times slower than Transformers, which constrains the scalability of RNNs.
- GNN: GNNs may not scale well with large and dynamic swarm networks, especially when temporal dynamics are involved.
- DQN/PPO: DQN and PPO require extensive online exploration and interactions (Levine et al., 2020), making them less practical for fault detection in complex systems with long-tail fault distributions.

A.5    ONLINE EVALUATION OF OUT-OF-DISTRIBUTION AND IN-DISTRIBUTION DATASET

| Test Region | APO (%) | APD (%) | HAR (%) | CAR (%) |
| --- | --- | --- | --- | --- |
| TR1 (OOD) | 20.0 | 26/34/21/19 | 10.8 | 29.7 |
| TR2 (OOD) | 31.5 | 46/32/11/11 | 4.9 | 64.1 |
| R4 (ID) | 27.3 | 16/29/29/26 | 6.5 | 48.7 |

Table 5: **Performance metrics of PM-2B .** The metrics include Action Probability per Observation (APO), Action Probability Distribution (APD), High-Value Action Ratio (HAR), and Constant-Pressure Action Ratio (CAR). Testing was conducted in three distinct regions: TR1 (rural, out-of-distribution), TR2 (urban, out-of-distribution), and R4 (suburban, in-distribution), to evaluate the model's generalization capability across diverse environments.

As is illustrated in Table. 5, the PM-2B model demonstrates strong generalization across different environments, maintaining high performance in both in-distribution (ID) and out-of-distribution (OOD) regions. In the OOD rural region (TR1 & TR2), the model achieves the comparable performance with ID region (in the context of comparing APO, HAR, and CAR). In contrast, the model's performance in the ID region (R4) shows more balanced APD values (16/29/29/26) than in OOD region, which could be a signal of overfitting.

A.6    ONLINE EVALUATION METRIC DETAILS

In this section, we provide a detailed explanation to selected metrics.

**Action Probability per Observation (APO)**    The definition of APO is

$$\text{APO} = \frac{\#\{\text{action generated as injected, testing method is called}\}}{\#\{\text{testing method is called}\}} \times 100\%,$$

where $\#\{\cdot\}$ denotes the number of occurrences of the specified event. APO aims to measure the percentage of times a testing method generates actions that are injected into the system, indicating how often the framework effectively targets the desired action space during testing. However, high APO may result in redundant action injections, as not all injected actions contribute to uncovering valuable information. Only critical actions that can reveal faults or vulnerabilities are truly significant for effective testing. Therefore, additional metrics about action quality and testing efficiency are necessary to evaluate the true effectiveness of the testing framework.

**Action Probability Distribution (APD)**   APD measures the proportion of different types of actions generated by the testing framework. It is represented as a vector indicating the percentage of each action type. A balanced APD ensures that the framework explores a diverse set of actions, while an unbalanced distribution may indicate bias toward specific types, potentially missing critical scenarios. Evaluating APD helps assess whether the testing method maintains comprehensive action coverage or if certain action types are underrepresented.

**Hazard Action Ratio (HAR)**   HAR is defined as

$$\text{HAR} = \frac{\#\{\text{actions result in return-to-go significantly raise comparing with summary}\}}{\#\{\text{injected actions}\}} \times 100\%,$$

where $\#\{\cdot\}$ denotes the number of occurrences of the specified event. In practice, we consider an action to be hazardous if the difference between *return-to-go* and the *summary* is greater than 0.4. his threshold indicates that the injected action has a substantial impact on the system, potentially leading to risky or unexpected outcomes. A high HAR reflects the framework's ability to generate high-risk scenarios, which is crucial for identifying critical vulnerabilities during testing.

**Constant-Pressure Action Ratio (CAR)**   CAR is defined as

$$\text{CAR} = \frac{\#\{\text{actions result in high return-to-go when summary is also high}\}}{\#\{\text{injected actions}\}} \times 100\%,$$

where $\#\{\cdot\}$ denotes the number of occurrences of the specified event. In practice, an action is categorized as constant-pressure if both the return-to-go and the summary exceed a threshold of 0.4. This indicates that the action consistently maintains a high level of risk or pressure in an already high-risk scenario. A high CAR shows that the testing framework is able to sustain pressure over a prolonged period, making it more effective at evaluating the resilience and stability of the system under stress.

**High Risk Scenarios per Million Flights (SPM)**   SPM measures the frequency of high-risk scenarios detected by the testing framework for every million simulated flights. A high SPM value indicates that the testing framework is effective in uncovering critical situations that pose potential threats to system safety. It helps quantify the robustness of the testing methodology in identifying rare but impactful scenarios.

**Faults per Million Flights (FPM)**   FPM represents the number of unique bugs identified for every million flights, where system may encounter severe failures. It reflects the framework's capability to discover actual system faults during testing. A higher FPM suggests that the testing strategy is not only triggering risky scenarios but also exposing underlying system vulnerabilities that need to be addressed before deployment.

## A.7   ARCHITECTURE AND TRAINING DETAILS

**Architectures of Policy Model**   The scenario-oriented testing framework for UTM systems consists of two main phases: training and inference (testing), as illustrated in Algorithms 1 and 2. Algorithm 1 details the training phase, where the Policy Model (PM) learns from an offline dataset of UTM scenarios. This phase involves iterating through epochs and batches, processing state-action-reward tuples, and updating the model parameters to minimize the prediction error for both actions and rewards. The training process incorporates context augmentation to enhance the model's ability to capture temporal dependencies. Algorithm 2 outlines the inference (testing) phase, where the trained PM is used to generate and evaluate potentially vulnerable scenarios in the System-Under-Test (SUT). This phase operates in a loop, continuously generating candidate actions, filtering them through an Action Sampler (AS), injecting selected actions into the SUT, and evaluating the outcomes. The process accumulates detected vulnerabilities while dynamically updating the context based on observed states, actions, and rewards. Together, these algorithms form a comprehensive approach to identifying potential faults and vulnerabilities in UTM systems, leveraging both historical data and adaptive, context-aware scenario generation.

---

**Algorithm 1** Training Phase of UTM Testing Framework

---

**Input:** Offline dataset $D$, Model architecture $M$
**Output:** Trained Policy Model PM
 1: Initialize PM with architecture $M$
 2: Initialize optimizer
 3: **for** each epoch **do**
 4:     **for** each batch $B$ in $D$ **do**
 5:         $s, a, r \leftarrow$ GetBatchData($B$)
 6:         $\tilde{s} \leftarrow$ AugmentWithContext($s$)
 7:         $\hat{a}, \hat{r} \leftarrow$ PM.Forward($\tilde{s}$)
 8:         $L \leftarrow$ ComputeLoss($\hat{a}, a, \hat{r}, r$)
 9:         BackpropagateAndUpdate(PM, $L$)
10:     **end for**
11: **end for**
        **return** PM

---

**Algorithm 2** Inference (Testing) Phase of UTM Testing Framework

---

**Input:** Trained Policy Model PM, System-Under-Test SUT, Action Sampler AS
**Output:** Detected vulnerabilities $V$
 1: Initialize vulnerability set $V \leftarrow \emptyset$
 2: Initialize context set $C \leftarrow \emptyset$
 3: **while** testing budget not exhausted **do**
 4:     $s \leftarrow$ GetCurrentState(SUT)
 5:     $\tilde{s} \leftarrow [C; s]$                                          ▷ Augment state with context
 6:     $R_{predicted} \leftarrow$ PM.PredictRTG($\tilde{s}$)
 7:     $a_{candidates}, \leftarrow$ PM.GenerateActions($\tilde{s}, R_{predicted}$)
 8:     $a_{filtered} \leftarrow$ AS.FilterActions($a_{candidates}$)
 9:     $a \leftarrow$ AS.SampleAction($a_{filtered}$)
10:     InjectAction(SUT, $a$)
11:     $R_{actual} \leftarrow$ EvaluateAction(SUT, $a$)
12:     **if** IsVulnerability($r_{actual}$) **then**
13:         $V \leftarrow V \cup \{(s, a, r_{actual})\}$
14:     **end if**
15:     UpdateContext($C, s, a, r_{actual}$)
16: **end whilereturn** $V$

---

|  | PM-1.2B | PM-2B |
|---|---|---|
| Layers | 64 | 64 |
| Model Dimension | 1280 | 1600 |
| Attention Heads | 20 | 25 |
| Activation Functions | GELU | |
| Positional Embeddings | Sinusoidal | |
| Optimizer | AdamW | |
| Peak Learning Rate | $8 \times 10^{-4}$ | $3 \times 10^{-4}$ |
| Learning Rate Schedule | 1000 steps warmup & cosine decay | |
| Batch Size | 512 | 256 |
| GPUs | 16 | |

Table 6: **Overview of the key hyperparameters of policy model.** We display settings for 1.2B and 2B models.

**Action Space**   Considering feasibility in implementation, we defined the action space of PM with 2 types of actions: (1) **O**ne-time physical actions and (2) short-**D**uration digital actions. As shown in Table 7, PM is also enabled to generate scenario configurations with different parameter settings.

| NAME | TYPE | DESCRIPTION | PARAMETERS |
|---|---|---|---|
| Wind | O | Winds with the exceeding magnitude | Speed, Direction |
| Obstacle | O | Obstacles appearing in UAVs' routes | Size, Location |
| Network Jitter | D | Temporary network disconnection | Time Duration |

Table 7: **Action types of policy model.** We consider three types of action for each agent. The **O** stands for **O**ne-time physical actions and **D** stands for short-**D**uration digital actions.

**Loss function**   We made use of model with decision transformer style which had out-standing in sparse reward tasks (Bhargava et al., 2023). In favor of regression of PM, a multi-objective loss function is introduced in training consisting of following aspects with configurable weights: *return-to-go* to model observation and causality, *action mask* to model world background knowledge and *action* to model decision.

## A.8   INDUSTRY LEVEL UAV SWARM SIMULATOR

The industry level UAV swarm simulator we applied is designed to create a digital twin of drone swarms for accurate analysis of both UTM system and UAVs' behaviors in real-world environments and interactions between natural environment and the whole system. Powered by a physics engine, the simulator closely replicates real-world physics. Additionally, the simulator incorporates hardware-in-the-loop by integrating actual UAV flight control systems, which adds to the accuracy. The simulator supports a variety of environmental configurations, including buildings, moving objects like balloons and birds, lighting conditions, and wind effects, etc. Backed by a dedicated support team, the system's reliability can be continuously improved.

## A.9   ENVIRONMENT DETAILS

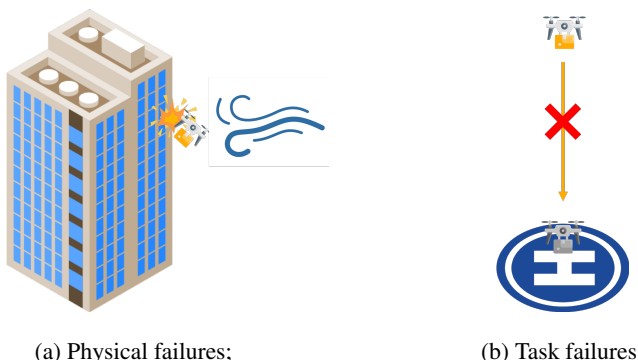

(a) Physical failures;  (b) Task failures;

Figure 7: **Two main types of failures in UTM.** Physical failures: Failures that result from physical damage or malfunction in system components, such as structural damage, hardware breakdowns, or external impact. These failures typically require immediate attention as they compromise the safety and integrity of the UAV or surrounding environment. Task Failures: Failures related to mission objectives, such as incorrect task execution, navigation errors, etc. Task failures impact the operational success and can disrupt planned missions or lead to unexpected behavior.

| Type | Index | Area | # of Airport | # of UAV | # of Flight Line | # of Alternate Airport | Fraction |
|------|-------|------|--------------|----------|------------------|------------------------|----------|
| Offline Training | R1 | Rural | 6 | 16 | 12 | 2 | 12.2% |
| Offline Training | R2 | Suburb | 12 | 24 | 24 | 7 | 18.3% |
| Offline Training | R3 | Urban | 6 | 36 | 18 | 6 | 27.5% |
| Offline Training | R4 | Suburb | 10 | 15 | 10 | 2 | 11.5% |
| Offline Training | R5 | Suburb | 10 | 15 | 10 | 2 | 9.2% |
| Offline Training | R6 | Urban | 8 | 16 | 16 | 2 | 12.2% |
| Offline Training | R7 | Urban | 4 | 12 | 8 | 3 | 9.1% |
| Online Testing | TR1 | Rural | 9 | 29 | 16 | 2 | N/A |
| Online Testing | TR2 | Urban | 6 | 16 | 16 | 6 | N/A |

Table 8: **Overview of training and testing regions used in the scenario-based testing framework.** Each region is categorized by type (rural, suburban, or urban) and is characterized by attributes such as the number of airports, UAVs, flight lines, and alternate airports. For training dataset, the fraction of each region is provided to reflect the distribution of different operational environments. Each region is specifically designed to provide a representative mix of operational challenges: regions R1 and R4 emphasize low-density rural and suburban operations, respectively, whereas regions R3 and R6 represent high-density urban areas with increased air traffic complexity. This distribution ensures the model learns to generalize across different environment types while prioritizing scenarios with a higher likelihood of critical interactions. Testing regions are designed to evaluate model performance on both trained dataset and unseen scenarios, ensuring robustness and generalizability.

