# OpenReview forum: "A Scalable Transformer-based Framework for Fault Detection in Mission-Critical Systems"
_ICLR.cc/2025/Conference — Submitted to ICLR 2025_

### Official Review · Reviewer_VBC2 · 2024-11-02

**Soundness:** 2
**Presentation:** 2
**Contribution:** 2
**Rating:** 5
**Confidence:** 3

**Summary:**

This paper presents a framework for detecting faults in Unmanned aircraft Traffic Management (UTM) systems using a Transformer-based architecture. The authors propose a scenario-oriented approach that combines a Policy Model (PM) with an Action Sampler (AS) to identify vulnerabilities in mission-critical systems. The work demonstrates improvements over traditional testing methods.

**Strengths:**

1. The paper tackles a critical problem: detecting rare, high-risk faults in Unmanned Aircraft System Traffic Management (UTM) systems.

2. Using a Transformer-based approach to capture long-term dependencies in system data is a smart choice, as the self-attention mechanism helps identify patterns over time.

3. The framework includes a preference bias and an Action Sampler to guide the model in selecting meaningful actions, reducing bias from the offline data.

**Weaknesses:**

1. The description of the state, action, and reward is too general, making it difficult to grasp the specific types of faults being tested and their impact on the UTM system. It would be helpful if the authors provided clearer examples or detailed scenarios, showing how particular actions (like network disruptions, obstacles, or drone malfunctions) are modeled and what consequences they produce in the system.

2. Some of the evaluation metrics are difficult to interpret. e.g., unclear definition of "bugs" in metrics.

3. Missing comparisons with other ML-based testing approaches (e.g., RNNs, LSTMs).

4. The accuracy shown in Figure 5 is relatively low (max ~10.8%), which may impact the reliability of the framework.

5. While the application of Transformers in UTM systems is interesting, the novelty of the approach feels limited, especially as the methods used (Transformer-based modeling, reinforcement learning, and scenario testing) are relatively well-established in similar problem spaces.

**Questions:**

1. Could you provide more details on this industry level simulator?
2. The paper references a "smoke test" as a baseline for comparison. Could you explain what this entails in the context of UTM testing?
3. Typically, validation sets are used for tuning hyperparameters, and the Figure 5 results should ideally be shown on test sets to evaluate the model's true generalization ability. Could you clarify/rename the exact setup here?

---

> ### Author Response · Authors · 2024-11-22
> **Response for VBC2 (1/5)**
>
> ## **For Weaknesses:**
>
> > (1) The description of the state, action, and reward is too general, making it difficult to grasp the specific types of faults being tested and their impact on the UTM system. It would be helpful if the authors provided clearer examples or detailed scenarios, showing how particular actions (like network disruptions, obstacles, or drone malfunctions) are modeled and what consequences they produce in the system.
>
> Thanks to your comments about formulation. We recognize the need for further specificity in describing the states, actions, and rewards in order to present a clearer picture of the types of failures involved in the test framework and their impact on the UTM system. To this end, we have improved the content in the paper through a multi-level narrative structure. Specifically, we add a detailed description of the UTM system functionality in Section 3, which explicitly describes the inputs and outputs of the UTM test framework and its relationship with the system interactions, in order to provide a clear context for the subsequent reinforcement learning modeling. In addition, in Appendix.1, we show the workflow of the UTM system architecture in detail through Figure 6, which explains how the testing framework generates targeted fault test scenarios based on real-time and historical data streams, while Table 5 lists the action space in detail, which covers a wide range of actions, including network outages, obstacle introduction, and UAVs experiencing high winds. From our practical experience demonstrated in Table 4, we can learn that these several representative action types are sufficient to create a sufficient number of anomaly scenarios through the selection and combination of injected objects. For example, network outages may lead to airspace congestion risks; obstacle introduction is used to evaluate the system's dynamic replanning capability under unexpected environmental constraints; and UAVs experiencing high winds test the system's robustness to cascading failures triggered by single- or multi-point failures. These specific examples are intended to clearly connect the framework from theoretical modeling to the actual fault detection process and highlight how the system reveals critical vulnerabilities through these tests.
>
> > (2) Some of the evaluation metrics are difficult to interpret. e.g., unclear definition of "bugs" in metrics.
>
> Thanks for pointing out wording issues. We improved expression by denoting *actions injected by testing framework* as `disturbances`  and *resulting UTM system states with abnormal UAV behaviors* as `faults` and `failures`  to clarify different concepts.
>
> > (3) Missing comparisons with other ML-based testing approaches (e.g., RNNs, LSTMs).
>
> Thank you for pointing out the need for comparisons with other ML-based testing approaches. We have addressed this concern by extending our discussion in Appendix.4, where we analyze the advantages of our Transformer-based framework compared to traditional models such as RNNs and LSTMs. Transformers inherently excel in capturing long-range dependencies due to their self-attention mechanism, which enables global context aggregation across input sequences. In contrast, RNNs and LSTMs, despite their proven ability to model sequential data, often suffer from vanishing gradient issues and limited temporal receptive fields, particularly when dealing with long trajectories or high-dimensional state spaces, as encountered in UTM fault testing. This limitation makes it challenging for RNNs and LSTMs to generalize effectively across diverse operational scenarios or capture complex temporal dependencies critical for fault detection. Transformer-based framework benefits from its ability to incorporate contextual information through an in-context learning approach, as discussed in Section 4.2. This capability allows the model to adapt to unseen scenarios more effectively, whereas RNNs and LSTMs are inherently less flexible due to their fixed architecture and reliance on shorter-term memory.

---

> ### Author Response · Authors · 2024-11-22
> **Response for VBC2 (2/5)**
>
> ## **For Weaknesses:**
>
> > (4) The accuracy shown in Figure 5 is relatively low (max ~10.8%), which may impact the reliability of the framework.
>
> We appreciate your concern regarding the action accuracy reported in **Figure 5** and its implications for the reliability of our framework. The accuracy values should be interpreted in the specific context of our framework's objectives and design rather than as standalone indicators of performance.
>
> The action accuracy metric in **Figure 5** reflects the **top-1 prediction accuracy** of the Policy Model (PM) when compared to ground truth actions derived from the offline dataset. It is worth noting that the framework's primary goal is not to perfectly reproduce the dataset's actions but to generate **high-value, targeted scenarios** that maximize fault detection efficiency [2]. In this sense, lower top-1 accuracy is not necessarily detrimental; it can indicate that the model is exploring novel or alternative scenarios rather than strictly adhering to historical patterns.
>
> Moreover, our framework incorporates **Top-K sampling** (Section 5.2), which leverages multiple plausible actions rather than solely relying on the highest-probability prediction. This approach enhances the diversity of generated scenarios and ensures comprehensive coverage of high-risk regions in the operational space. As demonstrated in the online evaluation results (Table 2), the framework achieves significant improvements in high-risk scenario discovery and fault detection efficiency, validating the practical reliability of the model despite the lower top-1 accuracy.
>
> We defined a set of targetted performance metrics in Table 1 and Appendix.6 in order to reveal performance of the proposed testing framework’s capabilities comprehensively from aspect of Action Balance, Action Quality and Testing Efficiency.
>
> Additionally, we observed a scaling effect in action quality as the model size increased, with larger models such as PM-2B consistently outperforming smaller ones in terms of fault discovery rates and robustness. This aligns with our hypothesis that the utility of the framework lies in its ability to uncover long-tail vulnerabilities rather than merely achieving high classification accuracy.

---

> ### Author Response · Authors · 2024-11-22
> **Response for VBC2 (3/5)**
>
> ## **For Weaknesses:**
>
> > (5) While the application of Transformers in UTM systems is interesting, the novelty of the approach feels limited, especially as the methods used (Transformer-based modeling, reinforcement learning, and scenario testing) are relatively well-established in similar problem spaces.
>
> We appreciate the your perspective on the perceived novelty of the approach and would like to clarify the unique contributions of our work within the context of established methodologies. While Transformers, reinforcement learning, and scenario testing are indeed well-recognized techniques, the integration of these methods in our framework addresses specific challenges that remain underexplored in the context of UTM fault detection.
>
> First, the novelty of our approach lies in how we formulate **fault detection in UTM systems** as a reinforcement learning problem tailored to address the long-tail distribution of fault scenarios. Unlike traditional RL or scenario testing methods, which often struggle with the sparsity and complexity of rare faults, our approach leverages **Transformer-based policy modeling** to dynamically capture inter-agent dependencies and temporal patterns in high-dimensional, multi-agent UTM systems. This is a departure from simpler, component-level testing frameworks, as it enables the identification of system-wide vulnerabilities that emerge from complex interactions over extended time horizons.
>
> Second, our work introduces a **context-aware testing framework** that combines offline RL with in-context learning to enhance generalization capabilities. By embedding historical trajectories into the state representation, our model dynamically adapts to unseen fault conditions, a feature not readily achievable with conventional testing frameworks or standard Transformer implementations. This mechanism bridges the gap between large-scale offline learning and real-time scenario generation, providing a scalable and adaptive approach to vulnerability discovery.
>
> Finally, our **domain-specific action sampling** methodology (detailed in Section 5.2) incorporates expert knowledge and safety constraints into the action space, ensuring the generation of physically plausible and operationally relevant scenarios. This integration not only refines the search space but also enhances the reliability and interpretability of the testing process—an aspect often overlooked in prior work.
>
> These contributions are supported by extensive evaluations in an **industry-level UTM simulation environment** (Section 6), where our approach demonstrated superior efficiency and effectiveness compared to traditional expert-guided methods and alternative ML-based baselines. By addressing the unique demands of fault detection in mission-critical systems, our work extends beyond the mere application of existing techniques, providing a tailored and practical solution to a challenging real-world problem.

---

> ### Author Response · Authors · 2024-11-22
> **Response for VBC2 (4/5)**
>
> ## **For Questions:**
>
> > (1) Could you provide more details on this industry level simulator?
>
> Thanks for your interest in simulator used in this research. We added a section as Appendix.8 to introduce this simulator, from which we had gathered >700h UAV fleet operation history trajectory data. This simulator targetted at system bahavior instead of on-device state only. As stated in Section 3.1 that the motivation of this research resided in detecting critical fault without realistic economic losses or human injury, utilization of simulator provided a workaround to test UTM. The simulator we applied features physics engine and hardware-in-the-loop for creating a digital twin of real-world scenarios.
>
> > (2) The paper references a "smoke test" as a baseline for comparison. Could you explain what this entails in the context of UTM testing?
>
> Thanks for pointing out concept issue about smoke test. We added explanation in caption of Table 2 (highlighted in blue) stating that smoke testing [1] referred to the basic functionality testing of UTM system, while expert-guided exploitation refers to the stress testing with expert knowledges.  Smoke test as a common developement stage, was conducted as the initial testing after a new build or version of the UTM system.
>
> > (3) Typically, validation sets are used for tuning hyperparameters, and the Figure 5 results should ideally be shown on test sets to evaluate the model's true generalization ability. Could you clarify/rename the exact setup here?
>
> Thank you for this important observation. We understand the need for clarity regarding the evaluation setup and agree that distinguishing between validation and test sets is critical for assessing the model’s true generalization ability.
>
> In our setup, the results presented in **Figure 5** correspond to evaluations conducted on **validation sets** that were not directly used for model training but were part of the same operational domains represented in the training data. These validation sets were utilized to monitor the training process, detect overfitting, and tune model hyperparameters. As the reviewer correctly pointed out, these results primarily reflect the model's performance on data similar to the training distribution, providing insights into learning stability and convergence behavior.
>
> To address concerns regarding generalization, we have also included results on **test sets** that represent **out-of-distribution (OOD) regions** (as described in **Appendix A.6**). These test sets are drawn from operational domains (TR1 and TR2) entirely excluded from the training process, offering a robust measure of the model's ability to adapt to new environments. This distinction is critical in validating the scalability and practical applicability of the proposed framework. The results on these test sets, detailed in **Table 2**, demonstrate the model's strong generalization capabilities, particularly in discovering high-risk scenarios and critical vulnerabilities beyond the training distribution.
>
> To avoid confusion, we improved the manuscript to explicitly label the results in **Figure 5** as validation performance and emphasized that generalization was rigorously evaluated on the OOD test sets. This clarification ensures a more accurate interpretation of the model's capabilities. Thank you for highlighting this opportunity to improve the manuscript's clarity.

---

> ### Author Response · Authors · 2024-11-22
> **Response for VBC2 (5/5)**
>
> ## Reference
>
> [1] Chauhan, Vinod Kumar. "Smoke testing." Int. J. Sci. Res. Publ 4.1 (2014): 2250-3153.
>
> [2] “Test efficiency: How it differs from Test Effectiveness?,” testsigma, https://testsigma.com/blog/test-efficiency/.

---

### Official Review · Reviewer_sDAK · 2024-11-03

**Soundness:** 1
**Presentation:** 2
**Contribution:** 2
**Rating:** 3
**Confidence:** 3

**Summary:**

This paper presents a novel Transformer-based framework for fault detection in mission-critical systems, specifically for Unmanned Aircraft System Traffic Management (UTM). The framework addresses fault detection in UTM by focusing on rare, high-risk failure scenarios, often missed by conventional testing techniques. Leveraging a Transformer-based policy model within a reinforcement learning (RL) framework, it generates high-risk scenarios. The approach includes a rule-based Action Sampler to enforce safety constraints, refined with human alignment, enhancing the realism of generated test scenarios. Empirical evaluations demonstrate an 8x improvement in vulnerability detection compared to traditional methods, with scalability confirmed by increasing the model size up to 2 billion parameters.

**Strengths:**

1. The problem is challenging and very important to the community.
2. The scenario-based framework seems promising.

**Weaknesses:**

1. The motivation of using transformer is unclear to me.
2. There is no formal problem statement and assumptions.
3. The evaluations are not sufficient to support the claims.

**Questions:**

1. I think it is important to clearly define what kinds of faults you are referring to. Do you mean a fault is a series of actions? If so, I don't understand why this framework is confined to MDP settings. If you are pointing to just the very first action, then it is very hard to determine whether it is a fault or not, maybe it is required to introduce some causality theory under this assumption.
2. I don't understand why a transformer-based architecture is required. Maybe it is better to include more insights and elaborate it with a motivating example?
3. Where would you deploy this model? I'm pretty sure this model cannot be deployed on each vehicle, and I'm not sure how this policy model can be updated in real-world applications.
4. I'm not entirely convinced by the experiments. First, the dataset is private and there is no evidence showing that faults are the 'long tails'. Therefore, it is hard to claim that the results support the claims in this paper. I think it is better to show some statistics about the dataset to help readers having a better understanding of it.

---

> ### Author Response · Authors · 2024-11-22
> **Response for sDAK (1/5)**
>
> ## **For Weaknesses:**
>
> > (1) The motivation of using transformer is unclear to me.
>
> Thank you for your feedback and for pointing out the need for a clearer explanation of the motivation behind using Transformers. We improved structure of Section 4 and added detailed discussion in Appendix.4 to make this motivation more explicit. The motivation for using Transformers in our framework is rooted in the unique challenges posed by fault detection in Unmanned Traffic Management (UTM) systems. Unlike simpler systems, UTM involves **multi-agent interactions** and **complex temporal dependencies** across extended operational timelines. Traditional architectures like RNNs and LSTMs struggle to capture these relationships effectively due to their sequential nature, which often leads to limitations in handling long-range dependencies and suffers from vanishing gradient issues.
>
> Transformers, on the other hand, excel at modeling these **long-range interactions** due to their self-attention mechanism, which allows for **global context aggregation** across all inputs. This is particularly important for identifying subtle, system-wide faults that emerge from **distributed agent behaviors** and **rare edge-case scenarios**—key challenges in UTM fault detection. By leveraging Transformers, our framework can efficiently capture the intricate dependencies between UAVs and their environment, enabling the generation of highly targeted and realistic fault scenarios.
>
> Additionally, Transformers' scalability aligns well with the requirements of our large-scale UTM simulation environment. As shown in our experiments (Section 6), increasing the model size significantly improves fault detection efficiency and scenario diversity, demonstrating the practical benefits of this architecture. The context-aware capabilities we incorporated (Section 4.2) further enhance the generalization of Transformers, enabling effective adaptation to unseen operational conditions, a feature less feasible with traditional approaches.
>
> > (2) There is no formal problem statement and assumptions.
>
> Thank you for pointing out the need for a more formal problem statement and clearer articulation of assumptions. We have addressed this concern in the revised manuscript by both formalizing the problem in **Section 4.1** and providing additional context about UTM and testing framework in **Section 3** and **Appendix.1** to strengthen the foundation of our work.
>
> **Section 3** provides a comprehensive introduction to the **UTM system architecture** and the **testing framework's input-output relationships**, offering essential context and challenges for the problem. This section explains how the testing framework ingests data streams such as UAV telemetry and environmental conditions, transforming them into actionable states for fault detection. To further enhance clarity, **Appendix Figure 6** illustrates the dynamic interactions between the UTM system and the testing framework, visually linking the components and their roles in the process.
>
> In **Section 4.1**, we formulate the problem of fault detection in UTM systems as a **Reinforcement Learning (RL) problem**, where the goal is to maximize the identification of critical failure modes in a complex multi-agent environment. Formally:
>
> - The **state space** $\( S \)$ is constructed from concatenated UAV telemetry data and system-wide mission statuses over a fixed time window, capturing both temporal and cross-agent dependencies.
> - The **action space** $\( A \)$, detailed in **Table 5**, includes a range of fault injection operations such as network disruptions, UAV malfunctions, and obstacle placement.
> - The **reward function** $\( R : S \times A \to \mathbb{R} \)$ quantifies the impact of actions based on metrics like system stability, collision risk, and mission completion, guiding the policy to prioritize the discovery of critical faults.
> - The objective is to learn a policy $\( \pi \)$ that maximizes the cumulative reward $\( R_\tau = \sum_{t=1}^T r(s_t, a_t) \)$, where $\( \tau \)$ is a trajectory of states and actions.
>
> We have also articulated key **assumptions** underlying the framework:
>
> 1. The UTM system exhibits a long-tail distribution of fault scenarios, with most cases being safe but rare edge cases posing significant risks, according to added **Table 3** and **Table 4** listing empirical knowledge of UTM fault distribution and detection.
> 2. The offline dataset used for RL is assumed to be sufficiently diverse to support robust policy learning.
> 3. Actions generated by the framework are constrained by domain-specific rules (enforced through the **Action Sampler**) to ensure physical plausibility and operational relevance.
>
> These additions serve to bridge theoretical definitions with practical implementation details, address the reviewer's concern and enhance the overall rigor of the manuscript.

---

> ### Author Response · Authors · 2024-11-22
> **Response for sDAK (2/5)**
>
> ## **For Weaknesses:**
>
> > (3) The evaluations are not sufficient to support the claims.
>
> Thank you for this important concern about the evaluation methodology. We would like to respectfully emphasize that our evaluation framework provides comprehensive validation of our claims. We have designed a rigorous set of metrics (detailed in Table 1 and Appendix A.6) that measure multiple aspects of performance, including action probability (APO, APD), action quality (HAR, CAR), and testing efficiency (SPM, FPM). These metrics provide quantitative evidence of our framework's effectiveness. In Section 6, we present extensive empirical analysis demonstrating not only the superior performance of our larger models but also a careful examination of overfitting phenomena through action accuracy and return-to-go loss measurements. Furthermore, in Appendix A.4, we provide detailed comparisons with alternative approaches like RNNs, GNNs, and DQN/PPO [1,2], highlighting why our Transformer-based approach better addresses the challenges of fault detection in UTM systems. Our framework has been validated across diverse environmental settings and successfully detected previously undiscovered failure modes. We believe this comprehensive evaluation framework provides robust empirical support for our claims about the effectiveness and scalability of our approach.

---

> ### Author Response · Authors · 2024-11-22
> **Response for sDAK (3/5)**
>
> ## **For Questions:**
>
> > (1) I think it is important to clearly define what kinds of faults you are referring to. Do you mean a fault is a series of actions? If so, I don't understand why this framework is confined to MDP settings. If you are pointing to just the very first action, then it is very hard to determine whether it is a fault or not, maybe it is required to introduce some causality theory under this assumption.
>
> Thank you for your insightful feedback. We appreciate your concern regarding the clarity of the term "faults" as used in our work, which has provided us an opportunity to refine the manuscript for greater precision and conceptual coherence.
>
> To clarify, the term "faults" in our framework does not solely refer to low-level failures at the UAV (machine) action level, but rather to emergent system-level anomalies in the Unmanned Traffic Management (UTM) system. Specifically, faults represent deviations from expected UTM behaviors, such as scheduling inefficiencies, airspace congestion, or unsafe operational states, which arise due to disturbances injected by the testing framework. These disturbances are deliberate modifications to the system environment or UAV states, designed to probe the robustness of the UTM system under adverse conditions.
>
> To address potential ambiguities, we have revised the manuscript to explicitly distinguish between `disturbances`—actions injected into the system for testing purposes—and `faults` or `failures`, which manifest as system-level aberrations in response to these disturbances. These distinctions are now clearly articulated in Sections 3 and 4 of the revised paper, ensuring a consistent and precise terminology throughout.
>
> Regarding the rationale for situating our framework within an MDP (Markov Decision Process) context, we emphasize that the faults targeted by our approach are emergent phenomena resulting from the cascading effects of disturbances over time. The MDP framework is particularly well-suited to model such temporal dependencies, as it captures the evolution of system states influenced by a sequence of actions. This allows us to effectively explore causal chains and identify vulnerabilities without necessitating additional causality theories.
>
> > (2) I don't understand why a transformer-based architecture is required. Maybe it is better to include more insights and elaborate it with a motivating example?
>
> Thank you for your concern about transformer. As we have stated in response to weakness 1, the transformer-based architecture is a natural fit for our framework because of its ability to model complex temporal dependencies and multi-agent interactions, which are critical in the UTM system. Faults in UTM often arise not from isolated events but from the cumulative effects of system-wide interactions over extended periods. Traditional approaches, such as simpler RNN-based or feedforward architectures, struggle to effectively capture these long-term dependencies and the high-dimensional state-action space of the system. Empirical results in Section 6 demonstrate that scaling our transformer model significantly improves fault detection efficiency compared to baseline methods, further validating its suitability for this task.

---

> ### Author Response · Authors · 2024-11-22
> **Response for sDAK (4/5)**
>
> ## **For Questions:**
>
> > (3) Where would you deploy this model? I'm pretty sure this model cannot be deployed on each vehicle, and I'm not sure how this policy model can be updated in real-world applications.
>
> Thank you for raising this important question about the deployment and application of our framework. We recognize that the distinction between the deployment environment and its intended role might not have been sufficiently clear in the original manuscript.
>
> To clarify, the proposed framework is **not designed for deployment on individual UAVs or other edge devices**. Instead, it is intended to be integrated into the development and testing pipeline of Unmanned Traffic Management (UTM) systems. After the model is trained offline, it serves as a tool for generating targeted test cases to evaluate the robustness of the UTM system. These test cases are designed to uncover system-level failures under diverse and challenging conditions, ensuring the UTM system's reliability prior to real-world deployment.
>
> The framework’s **policy model**, once trained, does not require real-time updates during deployment. It operates within the controlled environment of the UTM development process, where it generates high-risk scenarios for testing purposes. This approach circumvents the challenges of deploying computationally intensive models on resource-constrained devices, such as UAVs.
>
> In real-world applications, the policy model can be periodically retrained or fine-tuned using updated datasets derived from new UTM versions or observed operational data. This retraining process ensures that the framework remains effective as the UTM evolves to handle new use cases or unforeseen operational challenges[3].
>
> We have revised the manuscript to clarify these points in **Sections 3 and 5**, explicitly stating the intended deployment context and the role of the policy model in the UTM testing pipeline. We hope this resolves any misunderstanding and thank you for pointing out the need for greater clarity on this aspect.
>
> > (4) I'm not entirely convinced by the experiments. First, the dataset is private and there is no evidence showing that faults are the 'long tails'. Therefore, it is hard to claim that the results support the claims in this paper. I think it is better to show some statistics about the dataset to help readers having a better understanding of it.
>
> Thank you for your valuable feedback about dataset and experiments.
>
> To strengthen the validity of our claims about long-tail effect, we have made several updates to the manuscript. Specifically, we have added a more detailed description of the faults in real-world and UTM fault testing in **Table 3** and **Table 4**, where we summarize the types of faults encountered during testing and their distribution. These tables provide concrete statistics on the occurrence of faults, which will help readers understand the nature of the "long-tail" problems we aim to address. As is demonstrated in the Table, the real-world involves rare, high complex scenarios (e.g. multi disturbances on multi UAVs simultaneously or on single UAV sequentially) with lower than 0.1%, which is hard to generate during testing phase. However, once happened, these scenarios will may cost huge economic losses, or even human injuries/death.
>
> Additionally, in **Appendix 8**, we have included a comprehensive description of the **simulation environment**, including the scenarios, configurations, and fault injection strategies used to collect over **700 hours of real-world UTM operational data**. This dataset encompasses a range of operational conditions across various environments (e.g., urban, suburban, and rural areas), which we believe provides sufficient diversity and complexity to support our findings. These updates should help clarify how our experimental design aligns with the claim that faults in UTM systems exhibit a long-tail distribution and demonstrate that our approach can effectively identify and address these rare, critical failures.

---

> > ### Comment · Reviewer_sDAK · 2024-11-22
> >
> > Thanks for the clarification.
> > But this makes it even more confusing. The title is about fault detection and the authors state that it is for generating test cases for evaluating robustness? What are you really proposing? I encourage authors to revise it and make it clear.

---

> > > ### Author Response · Authors · 2024-11-22
> > > **Responses to further questions of reviewer sDAK**
> > >
> > > We thank the reviewer for your quick feedback and for raising concerns about potential confusion in our paper's terminology! For the definition of term "fault detection", we noticed that there could be a confusion with the process identifying faults during the operation of a system (what you may think it could be) and the **process of generating testing risky scenarios to detect possible faults in the pre-deployment testing phase** (what we truly proposed). We believe this definition is commonly used in the field of software testing [1] and software reliability [2], which may not cause a misunderstanding to the readers in the related fields.
> > > To further reduce any potential misunderstanding, we have added the **Section 3** and **Appendix A.1 and A.2** to better clarify our definition of terminology.
> > >
> > > **References**
> > >
> > > [1] Wong, W. Eric, et al. "Effect of test set minimization on fault detection effectiveness." Proceedings of the 17th international conference on Software engineering. 1995.
> > >
> > > [2] Kuo, Sy-Yen, Chin-Yu Huang, and Michael R. Lyu. "Framework for modeling software reliability, using various testing-efforts and fault-detection rates." IEEE Transactions on reliability 50.3 (2001): 310-320.

---

> ### Author Response · Authors · 2024-11-22
> **Response for sDAK (5/5)**
>
> ## References
>
> [1] J. Devlin, ‘Bert: Pre-training of deep bidirectional transformers for language understanding’, *arXiv*, 2018.
>
> [2] S. Levine, A. Kumar, G. Tucker, and J. Fu, ‘Offline reinforcement learning: Tutorial, review, and perspectives on open problems’, *arXiv*, 2020.
>
> [3] K. Spalas, ‘Towards the Unmanned Aerial Vehicle Traffic Management Systems (UTMs): Security Risks and Challenges’, *arXiv*. 2024.

---

### Official Review · Reviewer_kixY · 2024-11-03

**Soundness:** 3
**Presentation:** 3
**Contribution:** 3
**Rating:** 5
**Confidence:** 3

**Summary:**

This paper presents a promising application of Transformer-based reinforcement learning for UTM fault detection, with clear potential for improving safety in mission-critical UAV systems. The writing is clear and easy to follow. However, it seems to lack real-world validation, reproducibility, and practical deployment guidance. Also, it doesn't adequately address ethical issues for safe-critical applications. Addressing these concerns in a revision could substantially improve the work’s impact and relevance. If the authors follow the recommendations above and address the noted concerns, I would like to raise the score.

**Strengths:**

1. Well-designed Transformer-Based Fault Detection Framework for Unmanned Aircraft System Traffic Management (UTM):
This paper introduces an innovative Transformer-based framework specifically designed to detect faults in mission-critical Unmanned Aircraft System Traffic Management (UTM) environments. The paper proposes a scenario-oriented approach for generating fault scenarios using a Transformer-based policy model (PM) combined with an action sampler (AS) for safe, targeted testing. This approach leverages reinforcement learning in a high-dimensional, temporal environment.
 2. Scalability and Robust Empirical Validation:
The model’s scalability is demonstrated by training versions with up to 2B parameters. Empirical validation through 700 hours of simulations in a large-scale, industry-grade UTM environment is a strong point, providing evidence that the proposed framework could significantly outperform traditional testing methods by detecting critical faults more efficiently. Results indicate an 8x improvement in vulnerability detection compared to expert-guided testing and a clear scaling effect between model size and performance. Also, the larger model could better leverage more training data based on the experimental observations. This thorough evaluation across diverse UTM scenarios and environments underscores the potential utility of this approach in mission-critical settings.
 3. Rigorous Well-formulated Evaluation Metrics and Comprehensive Experiments:
The authors employ a diverse set of evaluation metrics—Action Probability per Observation (APO), Action Probability Distribution (APD), Hazard Action Ratio (HAR), and Bugs per Million Flights (BPM)—to measure the efficacy and robustness of the testing framework. This sounds very rigorous and well-formulated. Besides, comprehensive testing is conducted on in-distribution and out-of-distribution environments, adding further robustness to the study. The systematic inclusion of both offline and online evaluations also provides a more holistic view of model performance, strengthening the empirical validation.

**Weaknesses:**

1. Lack of realistic real-world empirical study: I am particularly interested in whether the authors could include a realistic or real-world case study to complement their evaluation. As presented, the submission’s evaluation appears to be conducted solely within simulated, controlled environments, which may not fully capture the complexities or variances that arise in real-world scenarios. Incorporating a case study that applies the system or approach in a real-world setting would significantly enhance the paper’s impact by demonstrating its practical viability and relevance. Additionally, I would encourage the authors to address the Sim2Real (simulation-to-reality) gap, which often poses a significant challenge when transferring solutions from simulated environments to real-world applications. Understanding how the proposed methods or systems perform outside controlled simulations, and identifying potential limitations or adaptations required for practical deployment, would add substantial value to the paper and its contribution to the field.

2. Deployment concern: When deploying to resource-constrained embedded hardware such as UAVs, in addition to the accuracy of the task itself, real-time constraints should also be considered. Although the authors claim that validation accuracy can be significantly improved on a larger model (PM-2B) compared to a small model or traditional methods, the corresponding model's inference time may be significantly extended, making it impossible to meet real-time constraints, which in turn limits the deployment availability on real systems. In this case, solution efficiency is very important. Can the author provide some potential ideas to solve this concern? This raises a question: the evaluation metrics introduced by the author do not seem to take into account the cost of model inference. If it is taken into account, will the observed scaling law still hold? Or will Sweetspot be a specific model size?

3. Ethical concerns: The authors have not fully explored the ethical considerations involved in safety-critical applications, which is a significant oversight given the potential dual-use nature of adversarial fault detection models. In safety-critical fields, such as autonomous vehicles or medical diagnostics, ensuring that models are used responsibly and safely is essential. The dual-use nature of these models implies that they could potentially be misused if adequate safeguards are not in place. I recommend that the authors discuss strategies for preventing misuse and promoting ethical, responsible deployment practices. Moreover, it would be beneficial if the paper included a more thorough discussion on potential defensive mechanisms designed to detect or mitigate faults and adversarial attacks during the model’s deployment. Such considerations could bring the paper in line with current safety standards in the field, emphasizing the importance of robustness and reliability under real-world conditions.

**Questions:**

1. I am curious about whether the authors could provide any realistic/real-world case study. In this submission, it seems that the evaluation is conducted entirely in simulated controlled environments.

2. The evaluation metrics introduced by the author do not seem to take into account the cost of model inference. If it is taken into account, will the observed scaling law still hold? Or will Sweetspot be a specific model size?

2. The authors do not adequately address the ethical issues of safe-critical applications. Given the dual-use potential of adversarial fault detection models, the authors should discuss strategies for preventing misuse and ensuring responsible deployment. Additionally, having more discussion / considering defensive mechanisms for detecting or mitigating issues during model deployment would be valuable to align with safety standards in the field.

**Details Of Ethics Concerns:**

There is no discussion of the ethical implications or potential safety risks associated with applying a fault detection model in a safe-critical environment. Given the dual-use potential of adversarial fault detection models, the authors should discuss strategies for preventing misuse and ensuring responsible deployment. Additionally, having more discussion / considering defensive mechanisms for detecting or mitigating issues during model deployment would be valuable to align with safety standards in the field.

---

> ### Author Response · Authors · 2024-11-22
> **Response for kixY (1/4)**
>
> ## **For Weaknesses:**
>
> > (1) Lack of realistic real-world empirical study: I am particularly interested in whether the authors could include a realistic or real-world case study to complement their evaluation. As presented, the submission’s evaluation appears to be conducted solely within simulated, controlled environments, which may not fully capture the complexities or variances that arise in real-world scenarios. Incorporating a case study that applies the system or approach in a real-world setting would significantly enhance the paper’s impact by demonstrating its practical viability and relevance. Additionally, I would encourage the authors to address the Sim2Real (simulation-to-reality) gap, which often poses a significant challenge when transferring solutions from simulated environments to real-world applications. Understanding how the proposed methods or systems perform outside controlled simulations, and identifying potential limitations or adaptations required for practical deployment, would add substantial value to the paper and its contribution to the field.
>
> Thank you for raising concerns about real-world validation. Our empirical analysis demonstrates systematic fault detection patterns across testing phases before UTM system is deployed in real-world services. As is illustrated in Figure 6, our testing framework aims to mimic the real-world disturbance in the simulators, aiming to generate the scenarios where the UTM system may fail. As shown in Table 3, even after multiple testing phases, approximately 0.1% of faults remain undetected, particularly in high-complexity scenarios. This aligns with our real-world UTM failure distribution data (Table 3 and 4), where safe flights comprise around 94% of operations, but critical failures still occur - especially in complex scenarios involving multiple UAVs or simultaneous disturbances (about 0.01% of cases). These findings underscore the importance of comprehensive testing for rare but critical failure modes. To bridge the simulation-reality gap [1], our testing framework (detailed in Appendix A.2) employs an industry-level simulator with hardware-in-the-loop integration and precise physical modeling. We maintain strict alignment between simulated and real capabilities - the testing framework operates on identical data streams fed to both UTM and simulator, focusing on external disturbances like wind and obstacles rather than internal UAV functionality. This design ensures that testing insights translate reliably to real-world deployments while providing a safe environment for exploring edge cases.
>
> > (2) Deployment concern: When deploying to resource-constrained embedded hardware such as UAVs, in addition to the accuracy of the task itself, real-time constraints should also be considered. Although the authors claim that validation accuracy can be significantly improved on a larger model (PM-2B) compared to a small model or traditional methods, the corresponding model's inference time may be significantly extended, making it impossible to meet real-time constraints, which in turn limits the deployment availability on real systems. In this case, solution efficiency is very important. Can the author provide some potential ideas to solve this concern? This raises a question: the evaluation metrics introduced by the author do not seem to take into account the cost of model inference. If it is taken into account, will the observed scaling law still hold? Or will Sweetspot be a specific model size?
>
> Thank you for raising concerns about deployment constraints. Our framework is designed specifically for centralized server-side fault detection during UTM development, not for on-device deployment (as illustrated in Figure 6). As described in Section 6.2 , since we emphasize more on the detection effectiveness both training and inference are conducted on data center hardware, where model size constraints are less critical.
> We do not monitor or interfere with on-device UAV states - our framework only injects external environmental factors and analyzes system-level behaviors. Individual UAV inference and control remain handled by their existing lightweight controllers.
> Regarding model size optimization, our analysis in Section 6.2 shows varying performance versus compute tradeoffs across different scales. While PM-2B currently shows the best detection capabilities, determining the optimal model size for different operational requirements remains an valuable open research question.

---

> ### Author Response · Authors · 2024-11-22
> **Response for kixY (2/4)**
>
> ## **For Weaknesses:**
>
> > (3) Ethical concerns: The authors have not fully explored the ethical considerations involved in safety-critical applications, which is a significant oversight given the potential dual-use nature of adversarial fault detection models. In safety-critical fields, such as autonomous vehicles or medical diagnostics, ensuring that models are used responsibly and safely is essential. The dual-use nature of these models implies that they could potentially be misused if adequate safeguards are not in place. I recommend that the authors discuss strategies for preventing misuse and promoting ethical, responsible deployment practices. Moreover, it would be beneficial if the paper included a more thorough discussion on potential defensive mechanisms designed to detect or mitigate faults and adversarial attacks during the model’s deployment. Such considerations could bring the paper in line with current safety standards in the field, emphasizing the importance of robustness and reliability under real-world conditions.
>
> Thank you for raising concerns about ethical implications and safety. Our research specifically focuses on pre-deployment fault detection rather than operational security or ethical deployment practices, which are out of the scope of this research. We use simulation precisely because testing these failure scenarios in real environments would be prohibitively inefficient, costly and risky.
> As shown in Table 3, real-world UTM failures, though rare (~0.1% for complex scenarios), can have severe consequences. Our simulator-based approach (detailed in Appendix A.2) allows us to identify and address these critical faults during development, preventing them from manifesting in realisticly deployed systems where they could cause actual economic loss or human injury.
> In Section 6.2, we demonstrate how our framework successfully detected previously unknown vulnerabilities that could have impacted safety in real deployments. This proactive identification of failure modes through simulation directly contributes to safer system development, aligning with the fundamental goal of preventing critical failures before real-world deployment.

---

> ### Author Response · Authors · 2024-11-22
> **Response for kixY (3/4)**
>
> ## **For Questions:**
>
> > I am curious about whether the authors could provide any realistic/real-world case study. In this submission, it seems that the evaluation is conducted entirely in simulated controlled environments.
>
> Thank you for asking about real-world validation. Our empirical understanding is based on extensive industry data, captured in Table 3's testing progression analysis. Despite thorough testing phases (module, integration, smoke, stress), approximately 0.1% of complex faults remain undetected through traditional methods, particularly in high-complexity scenarios where simple testing approaches prove inadequate.
>
> | **Fault Types** | **Module Testing** | **Integration Testing** | **Smoke Testing** | **Stress Testing** | **Fault Remaining** |
> | --- | --- | --- | --- | --- | --- |
> | Module Level | ~20 % | ~10 % | ~30 % | ~40 % | ~0.1 % |
> | Interface Level | ~10 % | ~20 % | ~30 % | ~40 % | ~0.1 % |
> | Running time | ~10 % | ~10 % | ~40 % | ~40 % | ~0.1 % |
> | **Scenario Complexity** | Simple | Simple | Medium | Medium | High |
>
> This finding aligns with our real-world UTM failure distribution data (Table 4). While most operations proceed safely (around 94%), critical failures emerge in specific patterns: single-disturbance cases (around 5%), multiple-UAV scenarios (around 1%), and complex failure combinations (around 0.01%). These rare but critical scenarios drive our testing priorities.
>
> | Types | Number of Influenced UAVs | Disturbance Times within 60s | Case Example | Real-World Ratio |
> | --- | --- | --- | --- | --- |
> | Safe Flight | 0 | 0 | N/A | ~94% |
> | Disturbances | 1 | 1 | Winds with exceeding magnitude | ~5% |
> | Disturbances | ≥2 | 1 (each) | Winds hit multiple UAVs | ~1% |
> | Disturbances | 1 | ≥2 | Winds hit twice with 60s interval | ~0.1% |
> | Disturbances | 1 | ≥2 (simultaneously) | Signal Loss when Winds hit | ~0.01% |
>
> Our framework bridges simulation and reality through careful design choices (Appendix A.2). The simulator integrates hardware-in-loop flight control systems and physically accurate environmental modeling. By maintaining identical visibility and capability constraints between simulated and real UTM systems, and focusing on external disturbances rather than internal UAV states, we ensure testing insights transfer reliably to production environments while enabling safe exploration of edge cases.
>
> > The evaluation metrics introduced by the author do not seem to take into account the cost of model inference. If it is taken into account, will the observed scaling law still hold? Or will Sweetspot be a specific model size?
>
> Thank you for raising questions about model inference costs. Our framework focuses on server-side testing during UTM development rather than on-device deployment. As described in Section 6.2, Transformer-based testing framework run on server instead of UAV device where computation time is not a critical constraint compared to fault detection effectiveness.
>
> The model's inference cost becomes negligible in the context of our testing timeline - identifying critical faults that occur in just ~0.1% of real-world operations (Table 3) justifies additional computation time during the development phase. Our priority is maximizing detection capability to prevent these rare but severe failures before system deployment.
>
> > The authors do not adequately address the ethical issues of safe-critical applications. Given the dual-use potential of adversarial fault detection models, the authors should discuss strategies for preventing misuse and ensuring responsible deployment. Additionally, having more discussion / considering defensive mechanisms for detecting or mitigating issues during model deployment would be valuable to align with safety standards in the field.
>
> Thank you for your feedback on the ethical and safety aspects of our research.
> Our study is specifically focused on identifying critical faults during the development phase using a high-fidelity simulator, as detailed in Appendix A.2. This proactive approach allows us to uncover and rectify potential vulnerabilities before real-world deployment, where failures could lead to significant economic losses or human injury. Section 6.2 demonstrates our framework's success in identifying previously unknown faults that could have compromised safety in operational systems.
>
> While we acknowledge the value of discussing defensive mechanisms and ethical deployment practices, these topics are beyond the scope of our current research. Our contribution lies in enhancing system safety through rigorous pre-deployment testing, which is a fundamental step in developing mission-critical systems.
>
> We hope this clarifies the focus and scope of our work and how it contributes to safer system development by identifying potential failure modes prior to real-world deployment.

---

> ### Author Response · Authors · 2024-11-22
> **Response for kixY (4/4)**
>
> ## Reference
>
> [1] K. Spalas, ‘Towards the Unmanned Aerial Vehicle Traffic Management Systems (UTMs): Security Risks and Challenges’, *arXiv*. 2024.

---

### Official Review · Reviewer_w7PB · 2024-11-04

**Soundness:** 3
**Presentation:** 3
**Contribution:** 3
**Rating:** 8
**Confidence:** 3

**Summary:**

The paper proposes a scenario-based framework to search for long-tail cases to accelerate the fault detection process. By leveraging a transformer-based policy to capture the dynamics of tested system from the offline dataset, the paper is able to achieves over 8 times more vulnerability discovery efficiency compared with traditional expert-guided random-walk exploitation.

**Strengths:**

(1) The paper is pretty novel to leverage decision transformers to model thescenario generation problems.
The integration of context-aware scenario generation improves the model's generalization capabilities, allowing it to adapt better to novel situations. The authors also demonstrate the scalability of their model, effectively showcasing performance improvements as they increase the model size up to 2 billion parameters.

(2) The extensive empirical evaluation over 700 hours in a simulated environment provides strong evidence for the framework's effectiveness. The reported results of over 8 times more vulnerability discovery efficiency compared to traditional methods underscore the framework's practical utility. I like the fact that it also employs a variety of metrics for both offline and online evaluations, providing a well-rounded assessment of the framework's performance.

**Weaknesses:**

(1) The paper mentions the importance of diversity in generated scenarios, however, how do you plan to evaluate the diversity of generated scenarios? The approach will be of less meaningful if the generated scenarios are too similar.

(2) Although the paper highlights generalization capabilities, it does not adequately discuss potential overfitting issues, particularly with the larger model. An analysis of the balance between model capacity and overfitting in various environments would strengthen the paper.

(3)  While the authors compare their approach with expert-guided testing, it would be useful to include comparisons with other recent state-of-the-art methods in the fault detection domain (see questions), to better situate the contributions of their framework.

**Questions:**

(1) my biggest concerns are the overfitting. How you ensured that the model does not overfit to the training scenarios, given the huge training datasets you are using? What strategies are in place to evaluate generalization to unseen environments?

(2) The random approach is pretty weak baselines, have you considered other scenario generation approach as well? Like black-box optimization, RL? Could you also clarify how the metrics chosen for evaluation were determined? Are there other metrics that could provide additional insights into the framework’s effectiveness?

(3) What are the possible fault injection operations? Are they all included in Table 5 in the Appendix?

---

> ### Author Response · Authors · 2024-11-22
> **Response for w7PB (1/3)**
>
> ## **For Weaknesses:**
>
> > (1) The paper mentions the importance of diversity in generated scenarios, however, how do you plan to evaluate the diversity of generated scenarios? The approach will be of less meaningful if the generated scenarios are too similar.
>
> Thank you for your valuable comment regarding the evaluation of scenario diversity.
>
> Directly evaluation of diversity of generated scenarios is challenging due to the high-dimensionality and continuous nature of state space (as the analysis we added in Section
>
> We introduced metric Action Probability Distribution (APD) to measure how actions of different types distributed, as a signal of system exploration and secnario diversity. APD was defined in Appendix.6. Balanced distributed actions indicated that less critical scenarios missed. In Section 6.2 and Appendix.5, APD of different models and different environments were shown in Table 2 and Table 5, where non of them presents inbalance. And models showed more balanced APD in in-distribution region R4 than out-of-distribution regions TR1/2 as an evidence of effectiveness of APD metric.
>
> > (2) Although the paper highlights generalization capabilities, it does not adequately discuss potential overfitting issues, particularly with the larger model. An analysis of the balance between model capacity and overfitting in various environments would strengthen the paper.
>
> Thank you for raising this important concern about overfitting analysis. We added explicit overfitting analysis in Section 6.1 and learning curves shown in Figure 5 that demonstrated all model met initial increase in accuracy followed by decline. PM-2B model showed better resistance to overfitting with less return-to-go loss and later onset of overfitting than smaller models.
>
> Furthermore, we compared PM-2B model performance between in-distribution and out-of-distribution regions shown in Table 5.
> | Test Region | APO (\%) | APD (\%) | HAR (\%) | CAR (\%) |
> | --- | --- | --- | --- | --- |
> | TR1 (OOD) | 20.0 | 26/34/21/19 | 10.8 | 29.7 |
> | TR2 (OOD) | 31.5 | 46/32/11/11 | 4.9 | 64.1 |
> | R4 (ID) | 27.3 | 16/29/29/26 | 6.5 | 48.7 |
>
> The model demonstrates robust generalization across ID/OOD regions, evidenced by:
>
> - Consistent HAR values between OOD (4.9-10.8%) and ID (6.5%) regions, showing stable risk detection;
> - Comparable APO ranges (20.0-31.5% OOD vs 27.3% ID), indicating consistent action generation;
> - Similar CAR patterns in complex scenarios (29.7-64.1% OOD vs 48.7% ID), demonstrating maintained pressure capabilities.
>
> > (3) While the authors compare their approach with expert-guided testing, it would be useful to include comparisons with other recent state-of-the-art methods in the fault detection domain (see questions), to better situate the contributions of their framework.
>
> Thank you for your suggestion regarding comparative analysis. As we stated in Section 3 and Appendix.2, UTM fault testing was faced with multiple technical challenges which we explored to tackle with Transformer-base adversarial generation. As shown in Table 3, traditional development pipeline detected trivial faults stage by stage with few but critical faults remained. We discussed about algorithms like DQN and PPO in Appendix.4, that they required extensive online exploration and interactions, making them less practical for fault detection in complex systems with long-tail fault distributions.

---

> ### Author Response · Authors · 2024-11-22
> **Response for w7PB (2/3)**
>
> ## **For Questions:**
>
> > (1) my biggest concerns are the overfitting. How you ensured that the model does not overfit to the training scenarios, given the huge training datasets you are using? What strategies are in place to evaluate generalization to unseen environments?
>
> Thank you for raising these critical concerns about overfitting risk and generalization capabilities. Our architectural design incorporates multiple safeguards against overfitting. The context-aware scenario generation mechanism described in Section 4.2 (highlighted in blue) allows the model to adapt to novel situations by leveraging relevant historical patterns. Additionally, the Action Sampler with domain constraints (Section 5.2) helps prevent the model from generating implausible scenarios, while our preference bias compensation ensures balanced exploration of rare scenarios. The quantitative results strongly support our model's generalization capabilities. As shown in Table 5, performance remains consistent across both in-distribution and out-of-distribution environments. (Also mentioned in response to weakness 2.)
>
> Our evaluation protocol is particularly rigorous in testing generalization. We conducted over 700 hours of testing in completely unseen environments, including regions TR1 and TR2 that feature distinct characteristics from our training data (see Table 8 for configuration differences). The testing spans rural-urban transitions and varied infrastructure densities. Furthermore, our analysis of model scaling reveals that larger models actually demonstrate improved generalization, with PM-2B showing later onset of overfitting (Figure 5) and more consistent performance across OOD environments.
>
> > (2) The random approach is pretty weak baselines, have you considered other scenario generation approach as well? Like black-box optimization, RL? Could you also clarify how the metrics chosen for evaluation were determined? Are there other metrics that could provide additional insights into the framework’s effectiveness?
>
> Thank you for your questions about baseline comparisons and evaluation metrics. As shown in Appendix.4, while we evaluated alternative architectures, each showed significant limitations - RNNs/LSTMs exhibited 10x slower training than Transformers for sub-10M parameter models and struggled with gradient vanishing [3], GNNs showed performance degradation with dynamic swarm sizes, and DQN/PPO approaches [4] proved sample inefficient for long-tail distributions. Our evaluation metrics were chosen based on industry standards [2] and FAA guidelines [1] for UTM testing. As detailed in Table 1 and Appendix.6, metrics were developed for a comprehensive evaluation of the proposed testing framework’s capabilities. APO and APD measure exploration effectiveness, while HAR and CAR evaluate scenario criticality. SPM and FPM directly measure vulnerability discovery capability, which is the direct goal of the testing phase.
>
> > (3) What are the possible fault injection operations? Are they all included in Table 5 in the Appendix?
>
> We decribed action space implented in current model in Appendix.7 and Table 7 (shifted after revision). This action space covers major common types of UTM failures. Together with cofigurable parameters, it provided better coverage of potential disturbances. We added our empirical knowledge of real-world UTM system failures distributions in Table 4 where combination of different simple types of disturbance may result in critial faults at a relatively low rate.

---

> ### Author Response · Authors · 2024-11-22
> **Response for w7PB (3/3)**
>
> ## References
>
> [1] Faa, ‘UTM Field Test (UFT) Final Report’, *FAA*. Federal Aviation Administration, Nov-2023.
>
> [2] A. Hamissi and A. Dhraief, ‘A Survey on the Unmanned Aircraft System Traffic Management’, *ACM Comput. Surv.*, vol. 56, no. 3, Oct. 2023.
>
> [3] J. Devlin, ‘Bert: Pre-training of deep bidirectional transformers for language understanding’, *arXiv preprint arXiv:1810. 04805*, 2018.
>
> [4] S. Levine, A. Kumar, G. Tucker, and J. Fu, ‘Offline reinforcement learning: Tutorial, review, and perspectives on open problems’, *arXiv preprint arXiv:2005. 01643*, 2020.

---

> ### Comment · Reviewer_w7PB · 2024-11-25
>
> Thanks for the rebuttal and additional experiments. I like the general idea of exploring large models in the scenario generation. I will willing to raise my score to 8 considering the scale of models and data in this paper.
>
>
> Besides, I found those paper that may be enlightening to your future works, in case you are not aware of them.
> 1, Learning to collide: An adaptive safety-critical scenarios generating method
> 2, Safety-Critical Scenario Generation Via Reinforcement Learning Based Editing
> 3, Scalable End-to-End Autonomous Vehicle Testing via Rare-event Simulation
> 4, RealGen: Retrieval Augmented Generation for Controllable Traffic Scenarios

---

> > ### Author Response · Authors · 2024-11-26
> >
> > We sincerely thank the reviewer for your valuable and insightful feedback. We greatly appreciate the opportunity to build upon the recommendations provided, as they are highly relevant and closely aligned with our work.

---

### Author Response · Authors · 2024-11-22
**Global Response**

We sincerely thank all reviewers for their constructive feedback. We have substantially revised our paper to address the concerns, with key changes highlighted in blue. Below we detail the major revisions:
1. Expanded and clarified problem definition and challenges
    * Added **Section 3 FAULT DETECTION PROBLEM IN TESTING PHASE** to explain concepts, targets and relationship of UTM and UTM testing frameworks.
    * Detailed challenges residing in UTM fault detection problem in **Section 3.2** and **Appendix.3**.
    * Further introduce UTM and fault detection in **Appendix.1 and 2**.
    * Permutated **Section 2** to discuss more-related multi-agent system testing first.
2. Improved motivation analysis and problem formulation
    * Expanded motivation and implementation of usage of Transformer in **Section 4.2** in order to tackle challenges of long-range dependency and multi-agent interaction. More detailed discussion about Transformer was added in **Appendix.4**, including motivation of applying Transformer and comparison between Transformer and other models (e.g., RNN, LSTM, etc.).
    * Refactored Section 4 to introduce our designs which addressed challenges of UTM fault testing mentioned in **Section 3** correspondingly.
    * Improved notations used in formulations for reinforcement learning in **Section 4**.
3. Added detailed information of UTM systems and testing frameworks in favor of researchers from different fields to better comprehensive target problem.
    - Added basic design purposes and working pipeline of UTM testing framework in **Appendix.2**.
    - Illustrated whole system architecture of UTM, testing framework and simulator in **Figure.6**.
    - Discussed the motivation and reason that this work is carried out in simulated environment in **Appendix.2**. Critial cost of potential faults in realistic world is usually untolerable.
4. Expanded more background information about UTM fault testing problem
    - Added empirical knowledge about fault detection ratios across different stage of developement in **Table 3** and UTM failures distribution in realistic world in **Table 4**.
    - Added information of simualtor used in this research in **Appendix.8**.
5. Improved wordings and praragraph structures.

We have made extensive revisions to address the reviewers' concerns and strengthen our paper's contributions. We believe these changes significantly improve the paper while maintaining its core focus on scalable fault detection for mission-critical UTM systems. We welcome any additional feedback and thank the reviewers for valuable suggestions that helped enhance this work.

---

### Meta-Review · Area_Chair_k3Jr · 2024-12-23

**Metareview:**

This paper is on fault detection for Unmanned Aircraft System Traffic Management (UTM). A Transformer model is proposed for generating high-risk scenarios.  The framework consists of a Policy Model (PM) and an Action Sampler (AS). The PM can manipulate environmental factors (e.g., placing obstacles) and modify the internal states of drones (e.g., simulating poor network connectivity). The testing scenarios are further refined by  by the rule-based AS which takes human preferences into account.

Strengths: ML techniques to improve safety and reliability is an emerging area. The paper reports better vulnerability discovery compared with traditional expert-guided random-walk exploitation.

Weaknesses: real-world validation, reproducibility, and practical deployment guidance; the formulation appears too general to appreciate actual faults being tested and their impact on the UTM system;  Missing comparisons with other ML-based testing approaches (e.g., RNNs, LSTMs);  somewhat unconvincing accuracy which may impact the reliability of the framework.

**Additional Comments On Reviewer Discussion:**

The concern around real-world validation, reproducibility, and practical deployment were brought up by multiple reviewers in different words. The authors responded on the use of an  industry-level simulator with hardware-in-the-loop integration and precise physical modeling, ensuring alignment between simulated and real capabilities. Given the sharp application focus of the paper around UTMs, an expectation of some real world validation is reasonable. At the end of the rebuttal phase, this point was not fully resolved. This could partly also be a presentation issue - the  formulation appears fairly general with reviewers unable to fully appreciate actual faults being tested and their impact on the UTM systems in reality.  Concerns were also raised on somewhat unconvincing accuracy which may impact the reliability of the framework. It is certainly possible that "lower top-1 accuracy is not necessarily detrimental" but this requires deeper analysis which was not offered convincingly enough to move multiple reviewers leaning towards rejection.

---

### Decision · Program_Chairs · 2025-01-22

Reject